# Clonal hematopoiesis related TET2 loss-of-function impedes IL1β-mediated epigenetic reprogramming in hematopoietic stem and progenitor cells

J. McClatchy[1,2,9], R. Strogantsev[3,9], E. Wolfe [1,2], H. Y. Lin[1,2],
M. Mohammadhosseini[1,2], B. A. Davis [1,2], C. Eden[1,2], D. Goldman[4,5],
W. H. Fleming[4,5], P. Conley[6], G. Wu[6], L. Cimmino [7], H. Mohammed[3,10] &
A. Agarwal [1,2,3,4,8,10] ✉

Clonal hematopoiesis (CH) is defined as a single hematopoietic stem/pro-genitor cell (HSPC) gaining selective advantage over a broader range of HSPCs. When linked to somatic mutations in myeloid malignancy-associated genes, such as TET2-mediated clonal hematopoiesis of indeterminate potential or CHIP, it represents increased risk for hematological malignancies and cardio-vascular disease. IL1β is elevated in patients with CHIP, however, its effect is not well understood. Here we show that IL1β promotes expansion of pro-inflammatory monocytes/macrophages, coinciding with a failure in the demethylation of lymphoid and erythroid lineage associated enhancers and transcription factor binding sites, in a mouse model of CHIP with hematopoietic-cell-specific deletion of *Tet2*. DNA-methylation is significantly lost in wild type HSPCs upon IL1β administration, which is resisted by *Tet2*-deficient HSPCs, and thus IL1β enhances the self-renewing ability of *Tet2*-deficient HSPCs by upregulating genes associated with self-renewal and by resisting demethylation of transcription factor binding sites related to term-inal differentiation. Using aged mouse models and human progenitors, we demonstrate that targeting IL1 signaling could represent an early intervention strategy in preleukemic disorders. In summary, our results show that *Tet2* is an important mediator of an IL1β-promoted epigenetic program to maintain the fine balance between self-renewal and lineage differentiation during hematopoiesis.

Clonal hematopoiesis (CH) occurs when a single hematopoietic stem/progenitor cell (HSPC) gains a selective advantage over other stem cells in bone marrow in the absence of hematological malignancy[1]. A more clinically useful subclassification of CH, clonal hematopoiesis of indeterminate potential (CHIP) is present in 10% of individuals over 60

years of age, and is defined as a > 2% variant allele frequency in the peripheral blood of a somatic mutation in a myeloid malignancy-associated gene. CHIP is a potent risk factor for the development of cardiovascular disease, pulmonary disease, type 2 diabetes, and hematological malignancies[2–4]. Further, CHIP is associated with

adverse outcomes even after the onset of these malignancies[5]. Thus, understanding the mechanisms by which premalignant HSPCs gain a fitness advantage will assist in designing better strategies for disease monitoring, prevention, and surveillance.

*TET2* loss-of-function mutations are one of the most common mutations in CHIP, present in approximately one-third of cases[2]. *Tet2*-deficient murine HSPCs are shown to have increased self-renewal potential[6]. Studies have identified inflammatory cytokines, including IL1β, IL6, and TNFα, are elevated in individuals and murine models with TET2 CH. These studies suggest that inflammatory cytokines exert a selective pressure to drive the aberrant myeloid expansion of pre-leukemic cells in vivo[7–9]. Further, inflammatory stressors such as gut microbiota and atherosclerosis have been shown to promote the myeloid expansion of TET2 CH[8,10]. Mechanistic work with IL6 has highlighted resistance to apoptosis as a driving factor in *Tet2*-KO HSPC's expansion[7]. Further, TNFα favors in vitro expansion through a mechanism likely related to reduced apoptosis in *TET2* mutant HSPCs compared with wild-type cells[11]. Here, we focus on IL1β-modeled chronic inflammation, as targeting IL1β signaling is a promising cancer therapeutic target[12–14] and improves outcomes of cardiac ischemia in murine models of CH[15]. To date, a study integrating how long-term exposure to IL1β-driven inflammation alters the relative fitness of *TET2*-mutated HSPCs over healthy HSPCs by changing their transcriptomic and epigenetic landscape remains elusive.

Using murine models with pan-hematopoietic specific *Tet2* deletion we show these mice represent major features of *Tet2* myeloproliferation under IL1β-mediated chronic inflammatory stress. Specifically, we demonstrate that IL1β potentiates the self-renewal ability of *Tet2*-KO HSPCs with increased myeloid bias and differentiation to *Tet2*-KO specific pro-inflammatory MHCII[+] macrophages. These findings are mechanistically supported using single-cell transcriptomic analysis and methylation profiling of progenitors. Importantly, targeting IL1 signaling reduces aberrant myeloid expansion and delays *Tet2*-mediated myeloproliferation in aged mice; thus, offering a strategy for early intervention for individuals with CHIP.

## Results

### IL1β exposure promotes expansion and myeloid bias of *Tet2*-KO cells with elevated proinflammatory Ly6c[hi] and reduced Ly6c[lo] monocytes/macrophages

IL1β is a potent pro-inflammatory cytokine elevated in human and murine *TET2* loss-of-function mutations-mediated CH[15–17]. To identify the impact of chronic IL1β exposure on the function and differentiation of HSPCs in TET2-mediated CH, we first used pan-hematopoietic *Tet2* loss-of-function mice (*Vav-Cre Tet2*[fl/fl], referred as *Tet2*-KO). We transplanted an equal number of lineage-negative progenitors derived from the bone marrow (BM) of *Tet2*-KO (CD45.2) and WT (CD45.1) mice into WT (CD45.1/45.2 HET) recipients and 3 weeks post-transplantation treated daily with or without IL1β for 15 weeks (Fig. 1a). Flow cytometric analysis of the peripheral blood (PB) was performed weekly. BM and spleen cells were analyzed at week five and fifteen (Supplementary Fig. 1a). Myeloid (CD45[+]CD11b[+]) cell frequency was elevated (~90% of the PB) 3 weeks post-transplantation, as lineage reconstitution is likely incomplete at that time due to pre-conditioning of recipient mice with irradiation. However, myeloid bias remained elevated in *Tet2*-KO relative to WT to a greater extent in mice treated with IL1β for 15 weeks, while myeloid frequency normalized in vehicle-treated mice (Fig. 1b). Although, IL1β promoted the myeloid expansion of both *Tet2*-KO and WT cells, the extent of the expansion is higher in *Tet2*-KO mice indicative of an additive effect of TET2 loss-of-function and IL1β. Specifically, at week 15 IL1β promoted the aberrant myeloid expansion of the *Tet2*-KO CD11b[+]Gr1[hi] (CD11b[+]Ly6c[hi]Ly6g[hi]) subset[8]. Notably, this increase occurred without a change in the frequency of CD11b[+]Gr1[lo] (CD11b[+]Ly6c[lo]Ly6g[lo]) cells in the PB or BM. Similarly, CD11b[+]Gr1[hi] trended towards greater expansion than CD11b[+]Gr1[lo] in the spleen in *Tet2*-

KO relative to WT mice treated with IL1β (Fig. 1c). These trends were observed within five weeks of treatment (Supplementary Fig. 2a). T cell frequency was reduced in *Tet2*-KO relative to WT in all conditions; however, B cell frequency reduction was IL1β dependent for the PB over time as well as BM and spleen at week 15 (Supplementary Fig. 2b).

To model the acquisition of somatic *TET2* mutations in hematopoietic cells, we used previously published ROSA26-M2rtTA mice[18], which expresses a doxycycline-inducible shRNA targeting *Tet2* (*shTet2*). Lineage-depleted *shTet2* BM cells were transplanted into WT CD45.1/45.2 recipients in competition with WT CD45.1 BM cells and treated intermittently with doxycycline and IL1β over 12 weeks (Fig. 1d). Doxycycline-induced knockdown of *Tet2* was monitored by GFP expression and remained reduced by ~80% at the endpoint by quantitative PCR (Supplementary Fig. 3a, b)[18]. Exposure to IL1β increased the frequency of GFP-expressing *shTet2* cells in the PB relative to the vehicle. IL1β also exhibited elevated frequency of *shTet2* myeloid cells, which reverses after removal of IL1β and reached over 98% of the PB after 4 weeks of daily stimulation (Fig. 1e, Supplementary Fig. 1b). As observed with *Tet2* KO mice (Supplementary Fig. 2b), T cells were reduced in frequency in the PB of *shTet2* cells independent of IL1β, whereas B cell frequency loss was IL1β dependent (Supplementary Fig. 3c). BM T and B cell frequency was also reduced within IL1β treated *shTet2* cells compared to vehicle, with splenic T Cells trending towards a reduction (Supplementary Fig. 3d). Neutrophils (CD11b[+]Ly6c[int]Ly6g[+]) and CD11b[+]Ly6c[hi]Ly6g[-] (Ly6c[hi]) monocytes/macrophages were elevated in the spleen and BM of *shTet2* mice relative to WT mice treated with IL1β. Interestingly two-weeks withdrawal of doxycycline led to the reversal of *shTet2* IL1β specific expansion of Ly6c[hi] monocyte/macrophages and suppression of CD11b[+]Ly6c[lo]Ly6g[-] (Ly6c[lo]) monocyte/macrophages. In the same comparison, neutrophils did not significantly reverse in the spleen or BM, suggesting their elevation results from already expanded *shTet2* progenitor subsets (Fig. 1f, Supplementary Fig. 3e).

To ascertain whether a difference in CD45 isoforms accounts for the enhanced reconstitution in competitive repopulation experiments[19], we transplanted BM progenitors from WT or *Tet2*-KO CD45.2 donors competing with CD45.1 WT donors into WT CD45.1 CD45.2 recipients. Consistent with previous studies *Tet2*-KO cells had a significant advantage of over WT CD45.2 cells after 4- and 8-weeks post transplantation (Supplementary Fig. 3f)[6]. Further, to eliminate the possibility that alterations within myeloid populations were dependent upon lethal irradiation of recipient mice, we treated *Tet2*-KO CD45.2 and WT CD45.2 (*Tet2*[fl/fl]) mice with IL1β for five weeks in a non-transplant setting (Supplementary Fig. 4a). Consistent with the above findings, IL1β treated *Tet2*-KO mice exhibited exaggerated splenomegaly and myeloid expansion overtime in the PB relative to IL1β treated WT mice (Supplementary Fig. 4b). IL1β dependent reduction of T and B cell frequency were observed again in *Tet2*-KO. Notably, lymphoid cell number was not reduced in the PB, suggesting lymphoid frequency reduction is a result of myeloid expansion not due to reduced lymphoid cell production. IL1β independent reduction in T cell frequency was not observed in *Tet2*-KO mice without transplantation, suggesting it requires transplantation-dependent factors, e.g. irradiation, and replicative reconstitutive stress (Supplementary Fig. 4c, d). The ratio of Ly6c[hi] to Ly6c[lo] monocytes/macrophages and the frequency of neutrophils were elevated in the BM of *Tet2*-KO relative to WT mice treated with IL1β (Supplementary Fig. 4e, f). Our findings suggest TET2 loss-of-function enhances Ly6c[hi] monocyte/macrophage and neutrophil production in response to pro-inflammatory stress.

### IL1β exposure selectively expands *Tet2*-KO HSPCs while promoting their myeloid bias

To determine whether chronic IL1β exposure impacts the differentiation and expansion of *Tet2*-KO, we next analyzed the *shTet2* competitive transplantation model described in Fig. 1d to delineate changes in

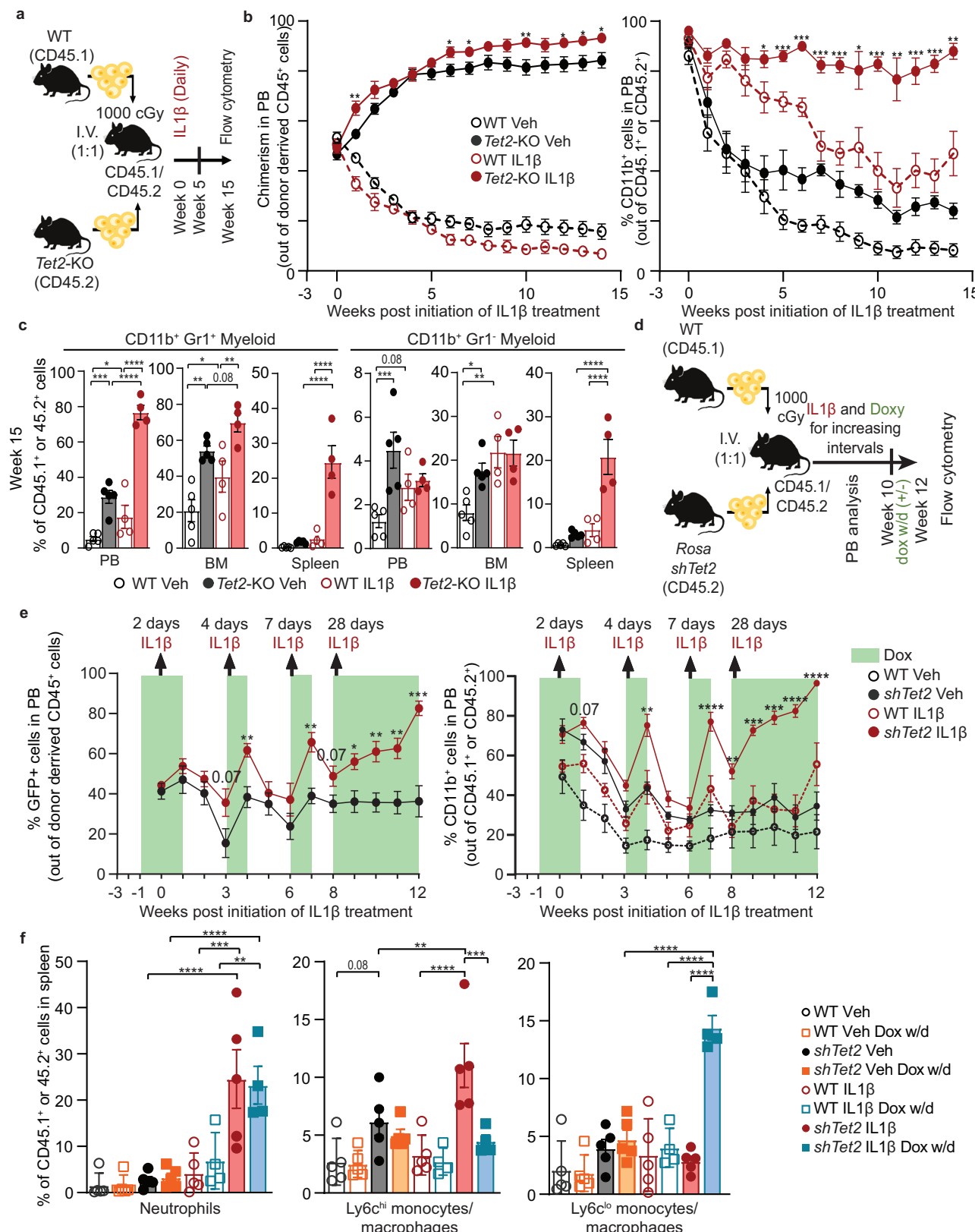

HSPCs populations. The frequency of *shTet2* LT-HSCs (Lin⁻c-Kit⁺Sca-1⁺Flk2⁻CD48⁻CD150⁺CD34⁻) for IL1β stimulated mice was greater than untreated *shTet2* or WT conditions in BM and spleens. This suggests IL1β drives expansion from the top of the hematopoietic hierarchy within LT-HSCs (Fig. 2a). Myeloid-erythroid-megakaryocyte biased multipotent progenitor 2 (MPP2s; Lin⁻c-Kit⁺Sca-1⁺Flk2⁻CD48⁺CD150⁺),

myeloid biased multipotent progenitor 3 (MPP3s; Lin⁻c-Kit⁺Sca-1⁺Flk2⁻CD48⁺CD150⁻) and lymphoid biased multipotent progenitor 4 (MPP4s; Lin⁻c-Kit⁺Sca-1⁺Flk2⁺CD48⁺CD150⁻) were all elevated in frequency in BM and spleens of *shTet2* relative to WT cells at steady state or upon IL1β stimulation (Fig. 2b). MPP3s exhibited the greatest fold increase within IL1β stimulated *shTet2* BM and spleens and their

**Fig. 1 | Chronic IL1β exposure enhances the expansion of *Tet2*-KO cells and myeloid bias with the elevation of Ly6c^hi^ over Ly6c^lo^ monocytes/macrophages.** **a**–**c** Lineage-depleted BM cells from wild-type (WT) CD45.1 and *Tet2*-KO CD45.2 mice were transplanted into WT CD45.1/2 mice which after 3 weeks were treated with IL1β or vehicle daily. **a** Experimental design, **b** percentage of WT CD45.1 or *Tet2*-KO CD45.2 cells out of all CD45⁺ cells (chimerism, significance shown between *Tet2*-KO with and without IL1β treatment) and the percentage WT CD45.1 and *Tet2*-KO CD45.2⁺ cells which are myeloid in the PB (significance shown between *Tet2*-KO and WT with IL1β treatment, n = 5 mice/group for Veh, 6 for IL1β) and **c** myeloid subsets (CD11b⁺Gr1^hi^, CD11b⁺Gr1^lo^) at week 15 (n = 5 mice/group for Veh, 4 for IL1β). **d**–**f** Lineage-depleted BM cells from WT CD45.1⁺ and Rosa-rtTA-driven inducible *Tet2* knockdown (*shTet2*) CD45.2 mice were transplanted into WT CD45.1⁺CD45.2⁺ mice, treated with doxycycline (dox) two weeks after transplantation and IL1β or vehicle three weeks after transplantation for increasing intervals up to 10 weeks and then with and without doxycycline (withdrawal; w/d) for an additional two weeks.

Arrows indicate the time relative to the initial treatment (week: 0, 3, 6, 8) and duration (days: 2, 4, 7, and 28) of daily treatment with IL1β or vehicle. Green indicates time and duration of doxycycline treatment. **d** Experimental design, **e** percentage of GFP⁺CD45.2 cells out of donor-derived CD45⁺ cells and the percentage of WT CD45.1 and *shTet2* CD45.2 cells which are myeloid (CD11b⁺) in the PB for doxy and IL1β or vehicle-treated experimental arms (significance shown between *shTet2* and WT with IL1β treatment, right panel, n = 5 mice/group), and **f** neutrophils (CD11b⁺Ly6c^mid^Ly6g⁺), Ly6c^hi^ monocytes/macrophages (CD11b⁺Ly6c^hi^Ly6g⁻), and Ly6c^lo^ monocytes/macrophages (CD11b⁺Ly6c^lo^Ly6g⁻) in the spleen with doxy or without doxy (w/d) for IL1β or vehicle treatments (n = 5 mice/group for Veh, IL1β, Dox w/d IL1β, 4 for Dox w/d mice/group). Error bars represent mean ± SEM. For panels c and f two-factor ANOVA determined family-wise error rate (FWER) adjusted *p* values. For panels b and e, a student's two-tailed t-Test determined the significance between single comparisons. For FWER adjusted and regular *p* values: *$p < 0.05$, **$p < 0.01$, ***$p < 0.001$, ****$p < 0.0001$.

---

reduction in frequency upon two weeks of doxycycline withdrawal was the greatest (Fig. 2c).

We validated these findings in a non-transplantation setting, and consistent with the previously published non-transplant models that examined 20 weeks or older *Tet2*-KO mice, LSK cells were elevated in the spleen upon IL1β stimulation, but not in BM (Supplementary Fig. 5a, b)[6,8]. In *Tet2*-KO mice, IL1β exhibited a significant increase in the frequency of MPP2s, MPP3s, and MPP4s in the spleen with the greatest increase relative to IL1β stimulated WT cells in MPP3s (Supplementary Fig. 5b).

Further down the hematopoietic hierarchy, common myeloid progenitors (CMPs; Lin⁻c-Kit⁺Sca-1⁺FCγRIII/II⁻CD34⁺) and granulocyte-monocyte progenitors (GMPs; Lin⁻c-Kit⁺Sca-1⁺FCγRIII/II⁺CD34⁺) were at higher frequency in the spleen of *Tet2*-KO mice relative to WT with IL1β stimulation but not at steady state. Notably, megakaryocyte-erythrocyte progenitors (MEPs; Lin⁻c-Kit⁺Sca-1⁺FCγRIII/II⁻CD34⁻) were similar in the BM and spleen of *Tet2*-KO and WT mice both at steady state and inflammatory conditions (Supplementary Fig. 5b). A higher myeloid bias in the spleens compared to BM in non-transplant settings is consistent with the previous finding of elevated extramedullary hematopoiesis in *Tet2*-KO mice[6,8], or may result from migration differences upon inflammatory stress.

To determine whether deletion of *Tet2* within GMPs is sufficient for peripheral blood expansion and myeloid biasing, we used *LysM-Cre Tet2^fl/fl^* (*LysM-Cre Tet2*-KO), whose promotor is active predominately within GMPs and downstream populations[20]. Equal numbers of lineage-negative progenitors derived from *LysM-Cre* (CD45.2) and WT (CD45.1) were transplanted into WT (CD45.1/45.2) mice and treated daily with IL1β stimulation for 18 weeks. No expansion or myeloid bias of *LysM-Cre Tet2*-KO occurred in the vehicle or IL1β stimulated mice in the peripheral blood as was observed in *Vav*-Cre and *shTet2* ROSA models (Supplementary Fig. 6). This demonstrates that *Tet2* deletion must occur at CMPs or upstream populations in the hematopoietic hierarchy for exaggerated myeloid bias and expansion to occur.

### Chronic IL1β exposure enhances the self-renewal capacity of *Tet2*-KO HSPCs driving their expansion

To assess how IL1β drives *Tet2*-KO HSPCs expansion in vivo, we treated WT and *Tet2*-KO mice with vehicle or IL1β for 5 weeks. We observed only a mild trend toward reduction in the frequency of apoptosis in HSPC subsets in *Tet2*-KO relative to WT mice (Supplementary Fig. 7a). We next examined the impact of IL1β-mediated stress on the cell cycle status of HSPCs in vivo. In a steady state, a significant portion of HSCs do not contribute to hematopoiesis and are largely quiescent, a mechanism to protect HSC function[21]. Upon in vivo IL1β administration, *Tet2*-KO LSKs trend towards a lower proportion of cells in S/G2M relative to IL1β-stimulated WT LSK cells (Supplementary Fig. 7b), suggesting a higher proportion of *Tet2*-KO LSKs maintained as $G_0/G_1$, although no significant difference was noted in either of those

populations. Given these findings and that *Tet2*-KO HSPCs are known to display increased self-renewal capacity at steady state[6,22], we tested whether IL1β stimulation cooperates with *TET2*-loss-of function to impact the self-renewal potential of HSPCs. Interestingly, IL1β significantly enhanced the self-renewal capacity of *Tet2*-KO LSKs relative to vehicle-treated *Tet2*-KO cells in a serial re-plating assay with increased CFU-GMs or CFU-Ms up to the quaternary plating (Fig. 2d). In contrast, WT LSKs were exhausted by the tertiary plating. Taken together, these studies suggest under prolonged IL1β induced inflammation, the primary fitness advantage of *Tet2*-KO HSPCs is mediated by increased self-renewal. Cumulatively these data alongside our findings in Fig. 2a–c suggest that IL1β-driven inflammation mediates HSC expansion predominately through increased self-renewal, with myeloid bias at the MPP stage of lineage commitment which is perpetuated at the CMP bifurcation for *Tet2*-KO mice (Fig. 2e).

### Single cell gene expression analysis reveals that *Tet2*-KO HSCs exhibit enrichment of transcriptional signatures associated with chronic stress, self-renewal capacity, and myeloid priming

To identify the mechanisms by which IL1β promotes myeloid expansion and self-renewal of *Tet2*-KO HSPCs, we performed single cell transcriptomic analysis of enriched lineage-negative BM cells derived from *Tet2*-KO and WT mice treated with vehicle or IL1β for 5 weeks (Fig. 3a). We identified 17 clusters both in WT and *Tet2*-KO mice (Supplementary Fig. 8a, Supplementary Data 1). Consistent with our flow cytometry analysis of *Tet2*-KO mice, IL-1β treated *Tet2*-KO mice showed an expansion of the granulocyte/monocyte progenitors cluster compared to respective wild-type controls (Fig. 3b). In order to identify changes that occur at the top of the hematopoietic hierarchy under inflammatory stress, we focused on the HSC cluster and identified substantially more (339 versus 41) upregulated differentially expressed genes (DEG) in *Tet2*-KO mice relative to WT treated with IL1β with little change in the number of down-regulated DEGs (Fig. 3c, d, Supplementary Data 2). Enrichr and STRING analyzes revealed enrichment for response to cytokines, stress, and reduced proliferation in *Tet2*-KO HSCs both at steady state and upon IL1β stimulation (Supplementary Data 3). Notably, *Tet2*-KO HSCs relative to WT enriched for transcriptional signatures of cancer, myelopoiesis, and dysregulation of lymphopoiesis and hemopoiesis upon IL1β stimulation but not in steady state (Supplementary Fig. 8b, c). Similar to gene ontology analyzes, gene set enrichment analysis (GSEA) of *Tet2*-KO relative to WT HSCs at steady state and under IL1β stimulation identified enrichment of many inflammatory pathways including IFNα/β, TNFα, and LPS regulated signaling as well as negative and positive mediators of MAPK signaling (Fig. 3e), suggesting *Tet2*-KO HSCs experience chronic inflammatory stress.

This inflammation signature includes phosphatases (*Ier2/5*, *Dusp1/2/5/6*), cell cycle inhibitors (*Btg2*) and transcription factors (*Egr1*, AP-1 members: *Fos*, *Fosb*, *Jun*, *JunD*; kruppel-like factor members: *Klf2/4/6*)

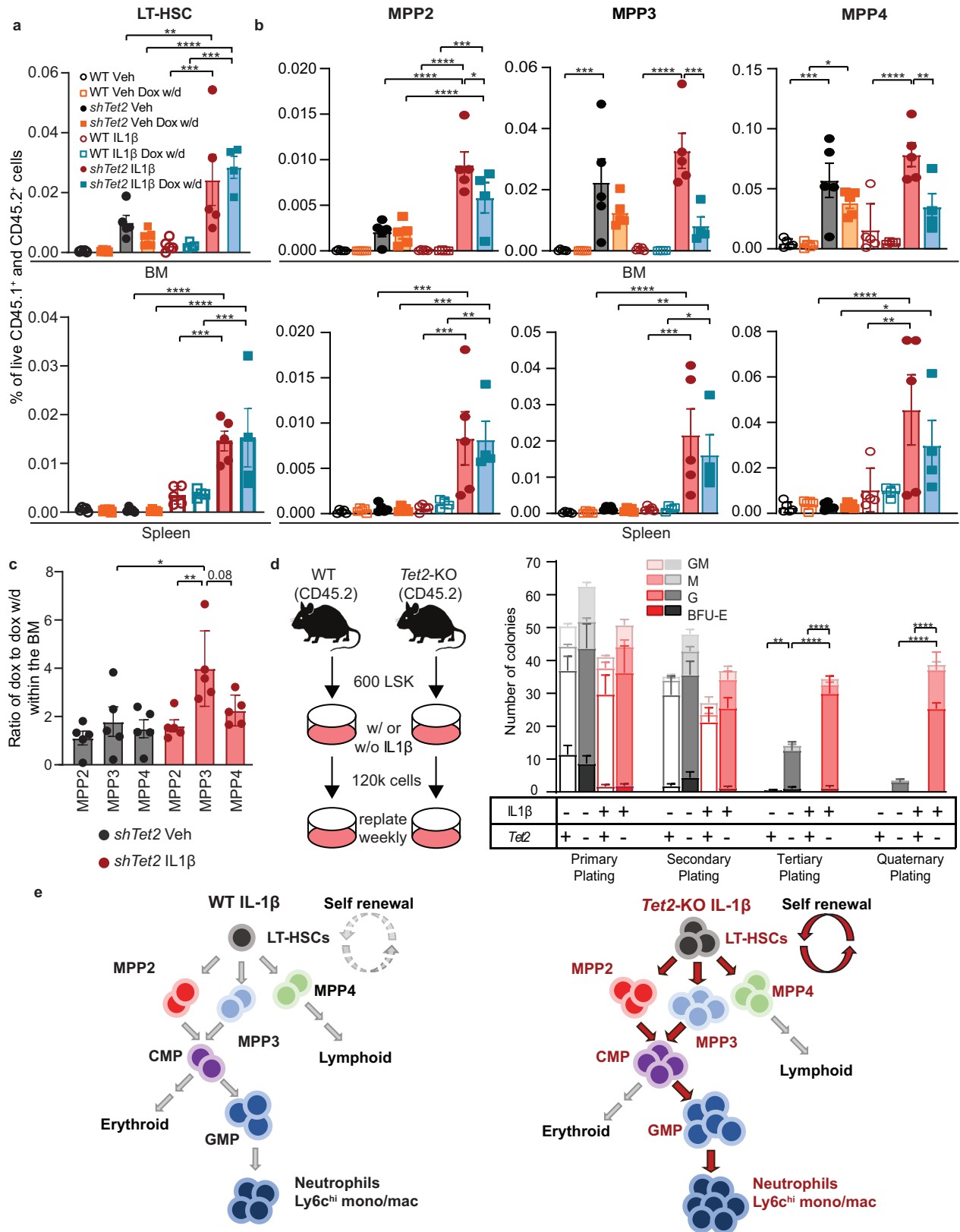

with roles in mediating and mitigating stress responses. Interestingly, in *Tet2*-KO relative to WT under IL1β stimulation, these inflammation and self-renewal associated DEGs are specific to the HSC cluster (Fig. 3f). Further, both at steady state and upon IL1β-mediated stress, several genes known to maintain HSC frequency under stress such as *Txnip* and *Kdm6b* were upregulated in *Tet2*-KO HSCs relative to WT[23,24]

(Supplementary Data 3). To query whether IL1β stimulation elicits similar stress responses to aging, we used a published old HSC gene signature[25]. This old HSC signature was enriched in *Tet2*-KO relative to WT HSCs at steady state and upon IL1β stimulation (Supplementary Fig. 8d). Cumulatively these results demonstrate *Tet2*-KO HSCs exhibit transcriptional signatures for chronic stress which can be magnified by

**Fig. 2 | IL1β drives expansion, myeloid bias, and enhances self-renewal of *Tet2*-KO HSPCs. a, b** Lineage-depleted BM cells from wild-type (WT) CD45.1⁺ and *Rosa-rtTA*-driven inducible *Tet2* knockdown (*shTet2*) CD45.2 mice transplanted into lethally irradiated WT CD45.1⁺CD45.2⁺ mice and treated with IL1β or vehicle as well as with and without doxycycline (withdrawal; dox w/d) as described in Fig. 1d and analyzed by flow cytometry (*n* = 5 for Veh, IL1β, Dox w/d IL1β, 4 for Dox w/d mice/group). **a** Percentage of live cells which are LT-HSCs and **b** MPP2, MPP3, and MPP4 in the BM and spleen of *shTet2* and WT mice. **c** The ratio of the change in frequency between dox to dox w/d for MPP2s, MPP3s, and MPP4s in BM. **d** Colony-forming unit (CFU) assay of sorted LSK cells from WT or *Tet2*-KO BM (*n* = 9 technical

replicates with 3 biological replicates); colonies were counted and the cells were re-plated every 7 days. M, macrophage; G, granulocyte; GM, granulocyte/macrophage; BFU-E, erythroid. **e** Simplified hematopoietic hierarchy with red text representing subsets elevated in frequency in *Tet2*-KO relative to WT mice treated with IL1β with red arrows connecting the most elevated subsets. Error bars represent mean ± SEM. For panels, **a**, **b**, and **d**, two-factor ANOVA was used to determine FWER-adjusted p values. For panel **c**, one factor ANOVA was used to determine the FWER-adjusted *p* values. For all FWER adjusted *p* values: *$p < 0.05$, **$p < 0.01$, ***$p < 0.001$, ****$p < 0.0001$.

---

inflammation such as IL1β. These findings also suggest *Tet2*-KO HSCs are adapted to better tolerate inflammatory stress, better maintaining their HSC pool relative to WT.

Consistent with higher tolerance of inflammation within *Tet2*-KO HSCs, GSEA identified significant enrichment of multiple pathways associated with self-renewal capacity in *Tet2*-KO HSCs relative to WT upon IL1β stimulation, whereas only a single pathway was significant at steady state (Fig. 3e). *Tet2*-KO HSCs relative to WT HSCs upon IL1β stimulation showed enrichment of HSC and leukemic stem cell signatures versus that of multipotent progenitor signatures. IL1β-stimulated *Tet2*-KO HSCs relative to WT counterparts showed predominant upregulation of genes associated with "self-renewal only" defined by GSEA analysis whereas genes associated with inflammation and self-renewal were identified in both vehicle and IL1β-stimulated cells (Fig. 3e, f). Specifically, we found significant upregulation of *Erg*, *Meis1*, and *Egr1* each associated with the self-renewal ability (Fig. 3f)[26–28]. Accordingly, proliferative signatures, which correspond to decreased self-renewal capacity, are downregulated in *Tet2*-KO HSCs relative to WT when treated with IL1β (Fig. 3e, Supplementary Fig. 8e)[22,29]. Cumulatively, these data suggest that *Tet2*-KO HSCs occupy a chronically stressed niche but display transcriptional programming that allows them to tolerate inflammatory insults by enhancing the transcription of self-renewal-promoting genes.

We found a myeloid leukemia signature was also enriched in *Tet2*-KO HSCs relative to WT at steady state and upon IL1β stimulation (Fig. 3e). Accordingly, genes with roles in myeloid biasing, such as *Hmga2* were upregulated with and without IL1β stimulation while *Hlf* was upregulated upon IL1β stimulation alone in *Tet2*-KO HSCs relative to WT (Supplementary Fig. 8e)[30,31]. Likewise, we identified upregulation in erythroid/megakaryocyte-promoting pathways. In contrast, pathways associated with lymphopoiesis were negatively enriched at steady state and upon IL1β stimulation, respectively (Fig. 3e, Supplementary Data 4). Accordingly, pseudotime analysis illustrated *Tet2*-KO IL1β treated mice exhibited increased frequency of myeloid lineage cells whereas erythroid and lymphoid lineages contracted after differentiating from HSCs and/or progenitors along pseudotime (Supplementary Fig. 9). Cumulatively, these findings support that IL1β promotes aberrant myeloid differentiation by impacting transcriptional programs in *Tet2*-KO HSCs.

To query whether the DEGs identified in *Tet2*-KO relative to WT HSCs at steady and pro-inflammatory states are also relevant to TET2 mediated-acute myeloid leukemia (AML) primary samples, we used a gene signature for vehicle and IL1β treated mice using the top 100 upregulated DEGs by significance in *Tet2*-KO relative to WT HSCs. Our murine HSCs-derived gene set was enriched in *TET2*-mutant AML relative to *TET2*-WT AML (Fig. 3g). Genes within the leading edge include the core inflammation and self-renewal signatures such as phosphatases (*DUSP1*), AP-1 members (*JUN*, *JUND*, *FOS*), and multiple other transcription factors (*KLF2*, *KLF6*, *EGR1*), suggesting these may also be relevant for TET2-dependent primary AML (Supplementary Fig. 8f).

### *Tet2*-KO HSPCs resist IL1β-driven methylation reduction of lineage-specific enhancer regions and terminal differentiation

### promoting transcription factor binding sites in a manner coinciding with cell fate

To identify DNA methylation patterns specific to *Tet2*-KO HSPCs, we performed whole genome bisulfite sequencing (WGBS) using purified LSKs, CMPs, and GMPs derived from the BM of *Tet2*-KO and WT mice treated for 5 weeks with and without IL1β (Supplementary Fig. 10a). As expected *Tet2*-KO LSKs, CMPs, and GMPs exhibited higher global methylation and a greater number of hypermethylated differentially methylated regions (DMR) than WT at steady state. These differences in hypermethylation between *Tet2*-KO and WT progenitors were exaggerated upon IL1β stimulation. This was largely resultant of the reduction of global methylation in WT LSKs, CMPs, and GMPs upon IL1β treatment (Fig. 4a, Supplementary Fig. 10b, Supplementary Data 5).

We found that DMRs in *Tet2*-KO LSKs, CMPs, and GMPs relative to WT were preferentially located in enhancer sites, with less within CpG islands, as described previously[32–34]. Notably, *Tet2*-KO LSKs relative to WT upon IL1β stimulation had the highest percentage of their probes hypermethylated for lymphoid enhancers, followed by myeloid, and erythroid enhancers. In contrast, the opposite pattern was observed in *Tet2*-KO CMPs relative to WT upon IL1β stimulation in which erythroid enhancers were hypermethylated most (Fig. 4b). This is consistent with our observation where upon IL1β stimulation, the differentiation of *Tet2*-KO HSPCs moved away from lymphoid-biased MPP4s. Similarly, CMPs biased away from megakaryocyte/erythroid committed MEPs (Fig. 2c, e) towards myeloid-committed GMPs.

To further validate this finding, we used published ChIP-seq data for 35 proteins including transcription factors (TF) and epigenetic modifiers with roles in the maintenance of HSC function, differentiation, or inflammation (Supplementary Data 6). The top 10 TF binding sites with the highest methylation increase in *Tet2*-KO relative to WT upon IL1β stimulation in CMPs included GATA1, TAL1, and MAZ which promote erythropoiesis (Fig. 4c)[35–37]. The latter two binding sites were in the top 10 for methylation increase at a steady state. Among these, the overrepresentation of MAZ motif was identified within hypermethylated DMRs by HOMER analysis (Supplementary Fig. 10c, Supplementary Data 7). In contrast, within LSKs many potent drivers of lymphopoiesis, including BACH2, BATF, ETS1, IKZF1, and RAG2 had binding sites which underwent pronounced hypermethylation in *Tet2*-KO relative to WT upon IL1β stimulation. HOMER analysis verified BACH2, BATF, and ETS1 were enriched in hypermethylated regions within LSKs but not in CMPs (Fig. 4c, Supplementary Fig. 10d). Cumulatively, these data suggest *Tet2*-KO HSCs lineage bias under inflammatory stress coincides with hypermethylation of different lineage-promoting enhancer and transcription factor binding sites throughout hematopoiesis.

Further, TFs known to drive terminal hematopoietic differentiation (ELF4, PU.1, CEBPB, JUN)[38–41] are among the most methylated binding sites in *Tet2*-KO LSKs upon IL1β stimulation. In contrast, these same TFs within CMPs are below the average change in methylation between *Tet2*-KO and WT for all genomic probes (Fig. 4c). These factors binding motifs were confirmed by HOMER analysis to be over-represented within DMRs in *Tet2*-KO relative to WT LSKs upon IL1β stimulation. Notably, these same factors were not enriched within CMPs, suggesting

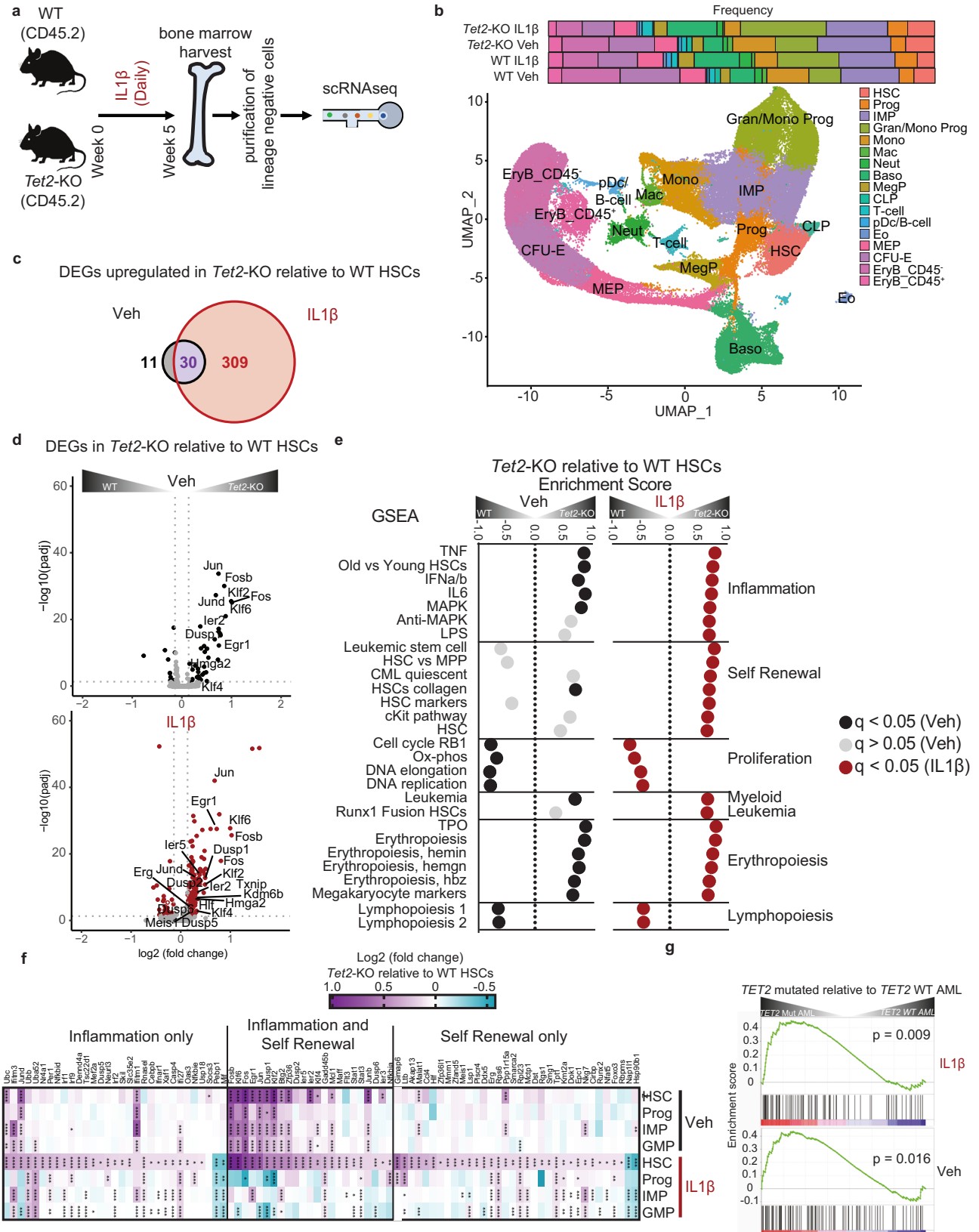

specificity to the LSKs (Fig. 4d). Thus, abrogation of transcriptional programs promoting terminal differentiation by the failure of TET2 to demethylate these binding sites may account for enhanced self-renewal capacity in *Tet2*-KO relative to WT under IL1β treatment.

We examined the top 150 downregulated genes from *Tet2*-KO HSCs relative to WT HSCs that are stimulated with IL1β which were

identified using single cell RNAseq analysis (Fig. 3). We found 38 genes proximal to hypermethylated binding sites of TFs (Fig. 4e). These include, promotor hypermethylation corresponding with regulators of proliferation (*Cdk4*, *Mki67*). Intragenic methylation differences were also prominent in many TFs binding sites which promote proliferation (*mki67*, *Rrm1*, *Rrm2*, and *Cdk4*), which may account for reduced *Tet2-*

**Fig. 3 | *Tet2*-KO HSCs exhibit transcriptional signatures for self-renewal capacity under inflammatory stress. a** WT and *Tet2*-KO mice were treated for 5 weeks with and without IL1β (n = 4 mice/group). BM cells were harvested and lineage-negative cells were enriched by magnetic selection. This enrichment increases the representation of progenitors but reduces the percentage of differentiated cells. Cells were analyzed by 10X single-cell RNA (scRNA) sequencing. **b** UMAP of 54,984 cells representing 17 clusters and bar graph representing their proportions including hematopoietic stem cell (HSC), progenitor (Prog), immature myeloid progenitor (IMP), granulocyte or monocyte progenitor (Gran/Mono Prog), monocyte (Mono), macrophage (Mac), neutrophil (Neut), basophil (Baso), megakaryocyte progenitor (MegP), common lymphoid progenitor (CLP), T cell (T-cell), plasmacytoid dendritic cells and B cell (pDc/B-cell), eosinophil (Eo), megakaryocyte-erythroid progenitor (MEP), colony-forming unit erythroid (CFU-E), erythroblast CD45⁻ EryB (EryB_CD45-) and erythroblast CD45⁺ EryB (EryB_CD45 + ). **c** Venn diagram of differentially expressed genes (DEGs) in *Tet2*-KO HSCs relative to WT treated with vehicle or IL1β. **d** Volcano plot of DEGs in *Tet2*-

relative to WT HSCs with or without IL1β stimulation. Selected genes based on enriched pathways are labeled. q values were determined using DESeq2. **e** Gene-set enrichment analysis (GSEA) from HSCs in *Tet2*-KO relative to WT showing selected significantly enriched gene sets (FDR < 0.05), with IL1β (red) or vehicle (black) or pathways not significantly enriched in the vehicle (grey). q values were determined by GSEA v4.2.1 using an empirical phenotype-based permutation test procedure. **f** Heatmap of upregulated DEGs in *Tet2*-KO relative to WT HSCs with and without IL1β stimulation for HSCs, Prog, IMPs, and GMPs. The genes shown are the lead genes for enriched pathways by GSEA analysis within HSCs, categorized into "Inflammation only", "Inflammation and Self-renewal", or "Self-renewal only" using the GSEA subcategories assigned in Fig. 3e. q values were determined by DESeq2. **g** Enrichment score plots for the gene signature of the top 100 upregulated genes in *Tet2*-KO relative to WT with IL1β (top) or vehicle (bottom) using DEGs from human *TET2* mutant (left) versus *TET2* WT (right) acute myeloid leukemia with p values determined by GSEA v4.2.1. For q values: *q < 0.05, **q < 0.01, ***q < 0.001, ****q < 0.0001.

KO HSCs proliferation in response to inflammatory signals, protecting their HSC pool (Supplementary Fig. 10e). Altogether, our data suggests that *Tet2*-KO HSCs lineage bias, self-renewal potential, and stem cell pool maintenance under inflammatory stress is mediated by differentiation state-specific epigenetic reprogramming.

### Genetic and pharmacological inhibition of *Il1r1* delays disease progression of *Tet2*-KO mice

To better determine how endogenous IL1β signaling through interleukin 1 receptor 1 (IL1R1) affects *Tet2*-mediated premalignant expansion in mice as they age, we bred *Tet2*-KO mice lacking *Il1r1* referred to as DKO (*Vav-cre Tet2ᶠˡ/ᶠˡIl1r1*) (Fig. 5a)[42]. We observed a positive correlation between age and the frequency of neutrophils and Ly6cʰⁱ monocytes in the PB of *Tet2*-KO mice, which was less pronounced in DKO mice, while Ly6cˡᵒ monocytes were not correlated to age in any condition (Supplementary Fig. 11a). At greater than 60 weeks of age, *Tet2*-KO mice display splenomegaly which trends towards reduction in DKO mice (Fig. 5b). Median survival for DKO mice is prolonged over *Tet2*-KO mice (median 71.86 versus 60.07 weeks, p = 0.07) (Fig. 5c). This is consistent with recent findings, in which genetic deletion of *Il1r1* has been implicated to rescue premalignant phenotypes of the *Tet2*-KO mice at 10 months (40.5 weeks) of age[43].

Ly6cʰⁱ monocyte/macrophages trended towards elevation within the BM. The majority of Ly6cʰⁱ monocytes/macrophages within the BM are F4/80⁺, suggesting they are macrophages (Supplementary Fig. 11b). These CD45⁺CD11b⁺F4/80⁺Ly6cʰⁱLy6g⁻ cells are MHCII⁺, and are likely the differentiated form of the recently identified *Tet2*-mutant specific MHCII⁺ monocyte population[44]. Interestingly, MHCII⁺ macrophages were elevated in frequency within the BM in *Tet2*-KO but not DKO with a similar trend within the spleen. *Il1r1*-KO alone also reduces MHCII⁺ macrophage frequency within the spleen relative to WT mice. F4/80⁻ Ly6cˡᵒ monocytes were reduced in frequency within *Tet2*-KO and DKO relative to WT and *Il1r1*-KO mice. F4/80⁻ Ly6cʰⁱ monocytes were reduced in the BM and trended towards elevation in the spleen in *Tet2*-KO relative to WT and DKO (Fig. 5d). This suggests that our previously observed Ly6cʰⁱ expansion within the spleen and BM primarily constituted Ly6cʰⁱ MHCII⁺ F4/80⁺ macrophages.

LSKs were elevated in the spleen of *Tet2*-KO relative to WT and DKO mice (Supplementary Fig. 11c) as has been observed previously[43]. Within MPPs, MPP3s trended towards overrepresentation in *Tet2*-KO, and MPP4s were underrepresented relative to WT mice, while these changes are less pronounced in DKO mice (Supplementary Fig. 11d). Cumulatively, these results suggest that the absence of *Il1r1* delays premalignant disease progression and mortality observed in *Tet2*-KO mice.

We next examined whether treatment with an FDA-approved IL1R1 antagonist anakinra[45] impedes *Tet2*-KO myeloid expansion in a competitive transplantation model (Fig. 5e). Mice treated with anakinra (Ana) alone or in combination with IL1β over 5 weeks suppressed

IL1β-mediated expansion of *Tet2*-KO cells in the PB (Fig. 5f, Supplementary Fig. 11e). Further, anakinra suppressed IL1β induced *Tet2*-KO myeloid expansion (Supplementary Fig. 11f). Similarly, anakinra reduced IL1β induced splenomegaly (Supplementary Fig. 11g). LK frequency was reduced with anakinra treatment in both WT and *Tet2*-KO mice upon IL1β stimulation; however, the effect is more pronounced in *Tet2*-KO mice. Further, anakinra treatment resulted in the reduced frequency of *Tet2*-KO LSK cells with or without exogenous IL1β stimulation, suggesting IL1R1 inhibition can reverse HSPCs similar to the baseline observed in WT mice (Fig. 5g).

Finally, we performed CRISPR-Cas9 editing of *TET2* within human BM progenitors. We noticed increased colony formation, which primarily constituted CFU-M, with *TET2*-edited progenitors upon IL1β stimulation but not with non-targeting controls. This effect on *TET2*-edited progenitors is reversed by IL1 receptor antagonist (IL1RA) (Fig. 5h, Supplementary Fig. 11h). These data support the efficacy of IL1 receptor blockade in abrogating IL1β-mediated expansion of *TET2*-mutated human HSPCs and myeloid cells.

## Discussion

Inflammation is widely implicated in promoting the expansion of TET2-mediated clonal hematopoiesis and associated myeloid bias[7–9,11]; however, the mechanism is not completely defined. We used multiple murine models of *TET2* loss-of-function to delineate the impact of IL1β-mediated inflammatory stress on hematopoiesis (Fig. 6).

We showed exogenous IL1β drives the expansion of *Tet2*-KO HSCs in vitro and in vivo. Elevated self-renewal capacity of *Tet2*-KO at a steady state, has been shown previously and is also observed by us[6,18]. Importantly, we discovered that IL1β substantially increased the self-renewal capacity of *Tet2*-KO LSKs relative to the steady state, while WT HSPCs were exhausted upon IL1β-mediated stress[22]. Accordingly, we identified the upregulation of transcriptional signatures associated with self-renewal potential in *Tet2*-KO HSCs predominately under IL1β stimulation. Notably, the binding sites of TFs known to drive HSCs towards terminal differentiation (PU.1, ELF4, JUN) were hypermethylated in a steady state in *Tet2*-KO relative to WT LSKs, which is further increased in magnitude upon IL1β stimulation. Similarly, IL1β promoted hypermethylation of CEBPB binding sites, known to drive HSCs towards terminal differentiation, and promoted hypermethylation in genes driving proliferation, such as *Cdk4*[38–41]. We found no significant differences in apoptosis between *Tet2*-KO and WT HSPCs upon IL1β stimulation, whereas reductions in apoptosis of *Tet2*-KO HSPCs have been shown upon chronic exposure to TNFα or acute exposure to IL6[7,11]. We note the majority of the axis dysregulated through IL6 (*Ptpn11*, *Morrbid*, *Bcl2l11*, *Bcl2*, *Casp1*) is not differentially expressed in the context of IL1β exposure, suggesting a different mechanism predominates[7]. In addition, Caiado et al. has recently shown that IL1α promotes CH through likewise mechanisms as IL1β[46]. However, it is

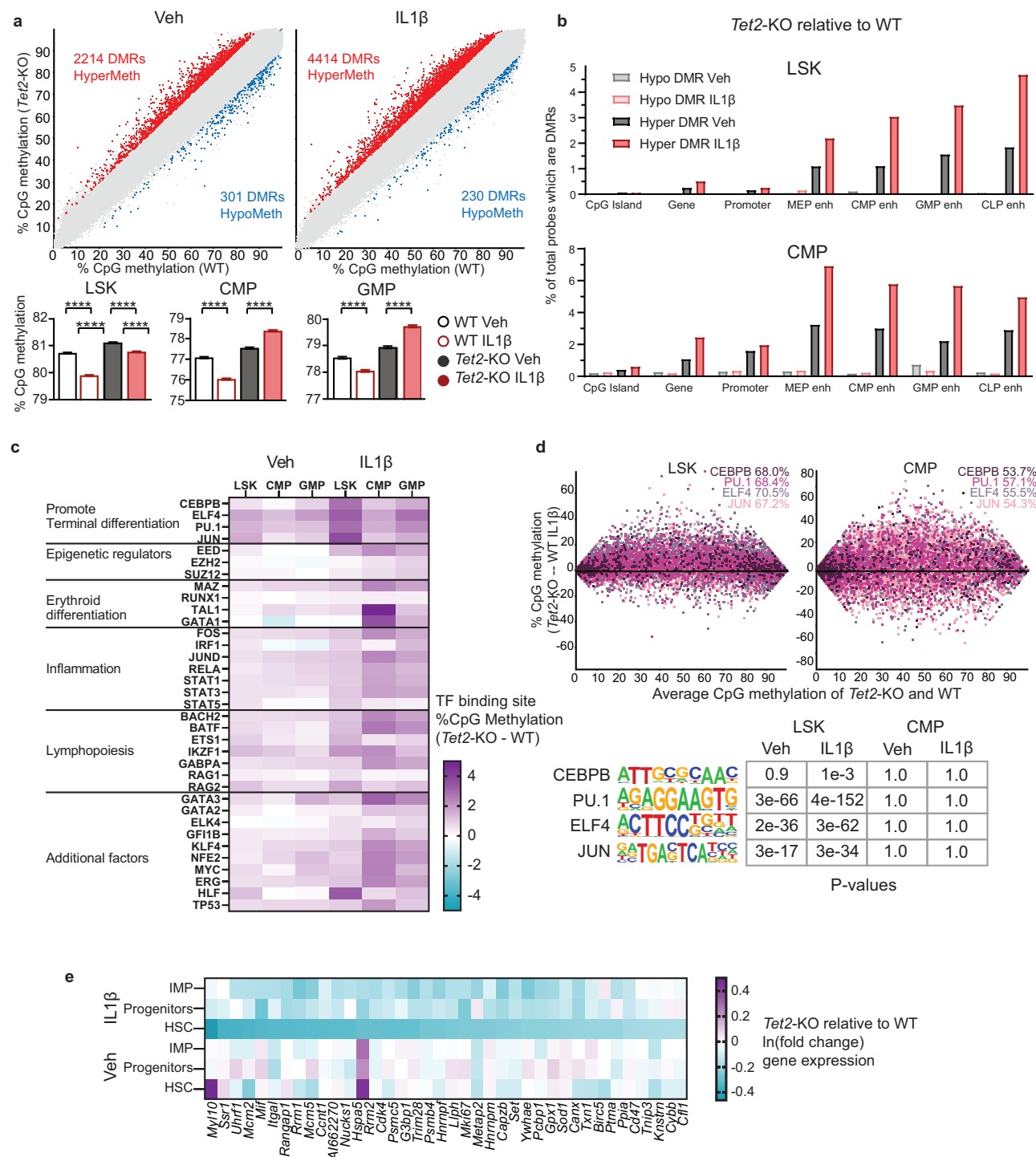

**Fig. 4 | *Tet2*-KO HSPCs resist IL1β-driven methylation reduction of lineage-specific enhancer regions and terminal differentiation promoting transcription factor binding sites.** WT and *Tet2*-KO mice were treated for 5 weeks with and without IL1β (*n* = 4 mice/group). BM cells were harvested and LSK, CMP, and GMPs were FACS purified and analyzed by whole genome bisulfite sequencing (Supplementary Fig. 10a). **a** Differentially methylated regions (DMRs) from 50 CpG long regions of the whole genome (>10% methylation difference, q < 0.05, statistical significance determined by SeqMonk) between *Tet2*-KO and WT LSKs with or without IL1β stimulation. The bar graphs below show the methylation level of all probes for each condition. **b** Percentage of 50 CpG segments overlapping each genomic element that are hypermethylated or hypomethylated DMRs in *Tet2*-KO relative to WT. **c** The percentage methylation difference between *Tet2*-KO and WT cells was analyzed using literature-derived ChIP-seq binding sites for each

transcription factor (TF). **d** Top panel, MA plots showing methylation difference for LSK and CMP cells derived from *Tet2*-KO and WT mice treated with IL1β using literature-derived ChIP-seq binding sites of terminal differentiation promoting TFs. The percentage of sites with higher methylation in *Tet2*-KO than WT are shown in the top right corner of each graph. Bottom panel, the statistical significance of motif enrichment determined by hypergeometric distribution (HOMER analysis tool) within DMRs for TFs with and without IL1β stimulation. **e** Heatmap of the 150 most downregulated genes near hypermethylated TF binding motifs in *Tet2*-KO HSCs relative to WT treated with and without IL1β. Their differential expression is shown across HSC, progenitors, and IMP clusters. Error bars represent mean ± SEM. For panel 4a (lower), two-factor ANOVA was used to determine the FWER adjusted *p* values: ****p < 0.0001.

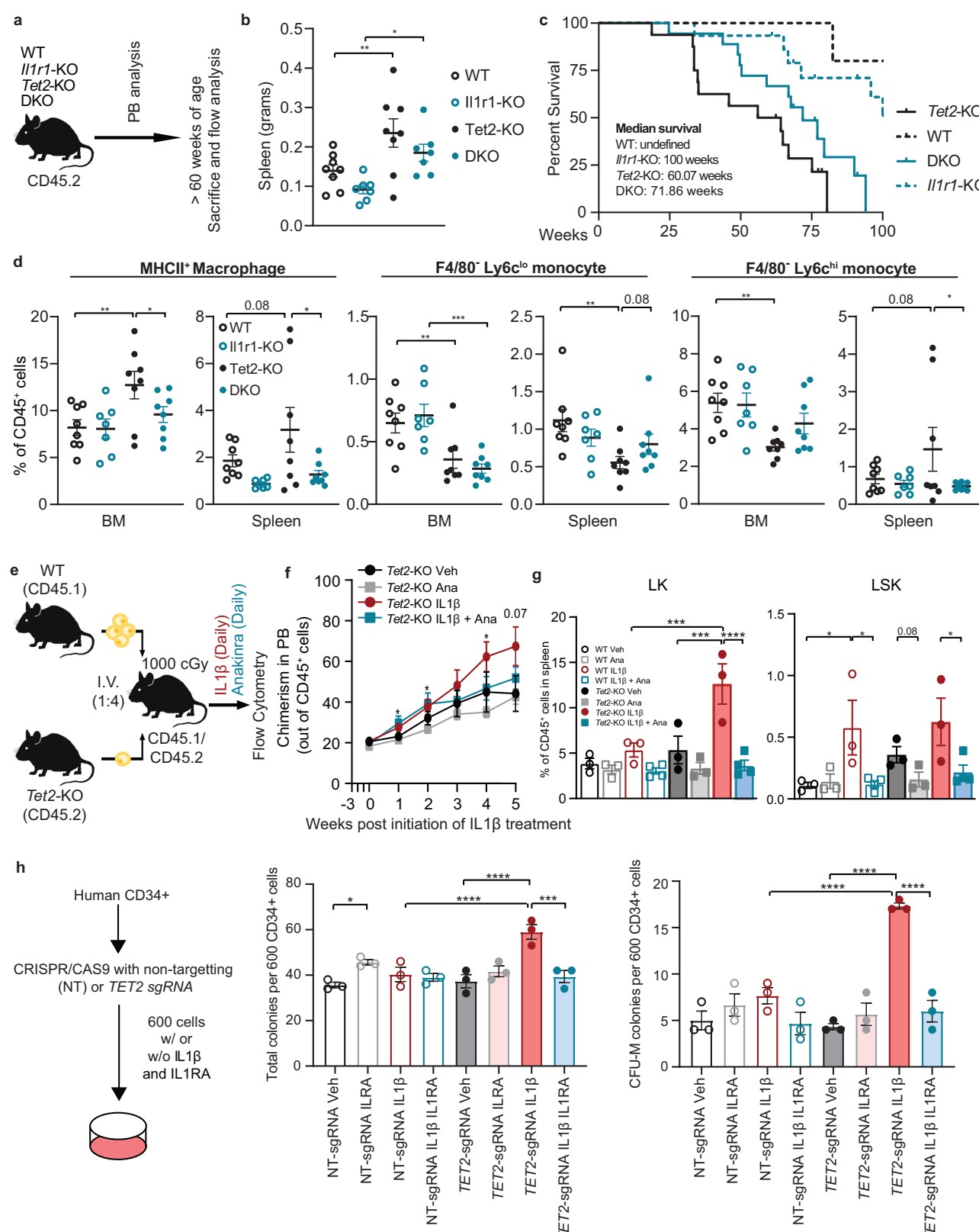

noteworthy that chronic IL1α increased proliferative signatures and reduced in vitro self-renewal capacity of *Tet2*[+/-] cells, whereas IL1β reduced proliferative signatures with increased self-renewal capacity *Tet2*-KO cells relative to WT counterparts. Whether this difference is resultant from homozygous versus heterozygous loss of *Tet2*, dosage dependencies, or differences between the spatially non-redundant roles of IL1β and IL1α remains to be investigated[47]. This suggests

various cytokines may differ in the mechanism by which they promote the expansion of *TET2*-mutated cells. Combined blocking of multiple cytokines or their downstream mechanisms may be necessary to fully abrogate clonal expansion.

Further, we identified a strong enrichment of the transcriptional signature specific to old HSCs[25] and AP-1 members (*Jun, Jund, Fos, Fosb*) which are downstream of IL1β and other pro-inflammatory signaling at

**Fig. 5 | Genetic and pharmacological inhibition of IL1R1 abrogates aberrant myeloid expansion in *Tet2*-KO mice. a–d** WT, *Vav-cre Tet2*$^{fl/fl}$ (*Tet2*-KO), *Il1r1-/-* (*Il1r1*-KO), and *Vav-cre Tet2*$^{fl/fl}$*Il1r1*$^{-/-}$ (double knockout; DKO) mice were bled twice and 8 mice were sacrificed at greater than 60 weeks of age, the remaining cohort was followed to observe survival outcomes. **a** Experimental design. **b** Spleen weight in grams. **c** Survival of WT (*n* = 8), *Tet2*-KO (*n* = 16), *Il1r1*-KO (*n* = 15), and DKO (*n* = 18) mice. **d** The frequency of MHCII$^+$ macrophages (CD45$^+$CD11b$^+$F4/80$^+$MHCII$^+$), F4/80$^-$Ly6c$^{hi}$ monocytes (CD45$^+$CD11b$^+$F4/80$^-$Ly6c$^+$Ly6g$^-$) and F4/80$^-$ Ly6c$^{lo}$ monocytes (CD45$^+$CD11b$^+$F4/80$^-$Ly6c$^-$Ly6g$^-$) in BM and spleen. **e–g** Lineage-depleted BM cells derived from WT CD45.1 and *Tet2*-KO CD45.2 mice were transplanted into lethally irradiated WT heterozygous CD45.1/2 mice. Three weeks after the transplantation mice were treated with IL1β (500 ng/mouse/day) or vehicle as well as with or without an IL1R1 antagonist (anakinra, 100 mg/kg) (*n* = 3 mice/group for Veh, Ana,

and IL1β, and 4 for IL1β Ana). **e** Experimental design, **f** the percentage of *Tet2*-KO CD45.2 donor cells out of all CD45$^+$ cells (chimerism) in PB, and **g** the frequency of LK and LSK cells in the spleen. **h** Human BM CD34$^+$ progenitors were CRISPR-Cas9 edited for *TET2* (*TET2*-sgRNA) alongside non-targeting control (NT-sgRNA), then 600 cells per well were plated with or without IL1β (25 ng/mL) and/or IL1RA (50 ng/ mL). Colonies were counted after 2 weeks and individual colonies were picked for sanger sequencing to determine indel frequency within *TET2*. Total colony numbers at 2 weeks and CFU-M (monocyte) colony numbers, with the number of colonies identified with detectable indels shown. Error bars represent mean ± SEM. For panels **a**, **d**, **g**, **f**, and **h** two-factor ANOVA was used to determine the FWER-adjusted *p* values. For panel **f**, a student's two-tailed *t*-Test was used to determine significance. For FWER adjusted and regular *p* values: *$p$ < 0.05, **$p$ < 0.01, ***$p$ < 0.001, ****$p$ < 0.0001.

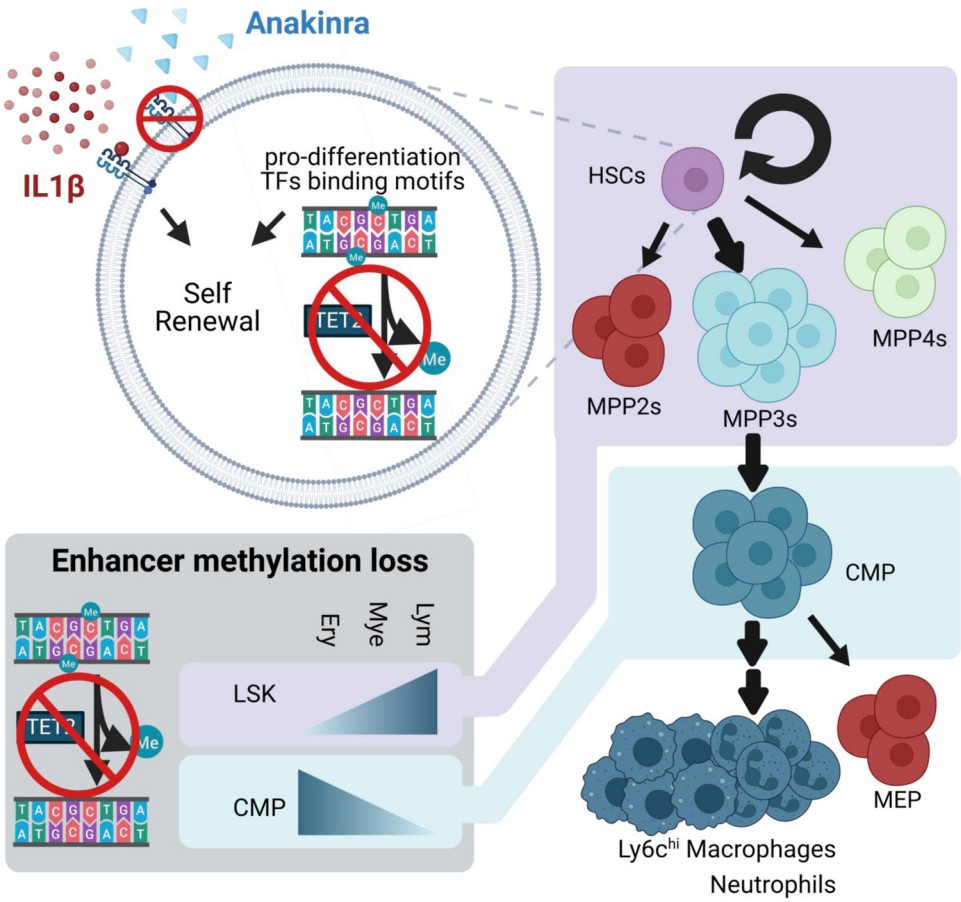

**Fig. 6 | Schematic depicting the impact of IL1β-driven inflammatory stress on hematopoiesis in *TET2*-mediated premalignancy.** Upon IL1β stimulation, *Tet2*-KO HSPCs showed enhanced self-renewal and bias towards MPP3s. Accordingly, *Tet2*-KO HSPCs (LSKs) resisted the loss of methylation observed in WT HSPCs upon IL1β stimulation within pro-differentiation transcription factors (TFs) binding motifs and lymphoid enhancers. Additionally, *Tet2*-KO HSCs exhibited enhanced transcriptional signatures for self-renewal and myeloid bias. Further downstream,

in the hematopoietic lineage, *Tet2*-KO CMPs also resisted methylation loss observed in WT CMPs within erythroid enhancers upon IL1β stimulation. Consistent with this GMPs frequency was elevated over MEPs in *Tet2*-KO relative to WT mice under inflammatory stress, leading to the expansion of Ly6c$^{hi}$ MHCII+ macrophages and neutrophils. Genetic or pharmacological inhibition of *IL1R1* signaling suppressed myeloid expansion and delayed premalignant phenotype over the course of aging. Created with BioRender.com.

steady state[48] and were exaggerated by IL1β stimulation in *Tet2*-KO HSCs relative to WT, suggesting *Tet2*-KO cells are primed to respond to pro-inflammatory stimulation. These pro-inflammatory genes were also observed in *TET2*-mutated versus *TET2* WT AML, suggesting human relevance, and they were specific to the HSC differentiation state, showing they are most pronounced in the population in which selective pressures are most relevant. However, *Tet2*-KO HSCs exhibited several putative mechanisms that may allow them to tolerate this continual stress. Predominantly under IL1β stimulation, negative regulators of AP-

1 (*Dusp1/2/5/6*) and NF-κB signaling (*Nfkbia*) are upregulated[49,50]. Further, there is hypermethylation of JUN binding sites in *Tet2*-KO LSK cells relative to WT in the steady state which is enhanced by IL1β stimulation. These changes were specific to LSKs and not observed in CMPs or GMPs. Cumulatively these data suggest *Tet2*-KO HSPCs perpetuate and tolerate a chronically inflamed microenvironment due to both transcriptomic and epigenetic reprogramming.

We discovered that in *Tet2*-KO relative to WT LSKs exhibited hypermethylation of lymphoid lineage-promoting TF binding sites at a

steady state which is enhanced by IL1β. This resulted from *Tet2*-KO resisting IL1β-induced demethylation of lymphoid TF binding sites observed in WT LSK cells and may explain why the lymphoid lineage is not elevated in frequency. This coincides with the IL1β induced bias for MPP3s within LSKs. In contrast, hypermethylation of erythroid TF binding sites and enhancers was elevated specifically within CMPs. After CMP bifurcation, there were increased GMPs over MEPs at steady state and upon IL1β stimulation in *Tet2*-KO relative to WT. We note that LSKs contain a heterogenous population and that MPP3s and GMPs are elevated in *Tet2*-KO mice relative to WT upon IL1β stimulation which may affect the prevalence of methylation in lineage-specific enhancers and transcription factors. Future studies may delineate methylation-specific effects within LSK, however, our findings are consistent with previous reports of transcriptional priming towards myeloid lineage with erythroid suppression at steady state[32]. Cumulatively these data suggest resistance to inflammation-mediated loss of methylation at enhancers and lineage-specific TF binding sites, coincides with cell fate decisions favoring myeloid bias upon *TET2*- loss-of-function.

IL1β elevated the frequency of neutrophils and Ly6c[hi] monocytes/macrophages in *Tet2*-KO relative to WT. Previous studies show that Gr1 (Ly6c/Ly6g) positive cells expand in *Tet2*-KO PB[8], spleen[6], and BM[51] upon inflammatory stress or aging and this is defined as aberrant myelopoiesis in response to inflammation[7,8,18,52]. Pro-inflammatory monocytes/macrophages have significant roles in cardiovascular disease[53] and are elevated in murine *Tet2*-KO models in steady state[4] which exhibit poor cardiovascular outcomes. Cardiovascular outcomes in these models are improved by IL1β inhibition[15]. In keeping with this, we identified that the expansion of the recently identified *Tet2*-KO specific MHCII+ macrophages[44] was largely IL1R1 dependent.

Recent studies show that *Il1r1* deletion abrogates *Tet2*-mediated premalignant disease including HSPC expansion in adult mice (10 months of age)[43]. In addition, we show that the absence of *Il1r1* prolonged *the* survival of *Tet2*-KO mice when followed for 23 months and the mitigation of MHCII[+] macrophage expansion. IL1 signaling blockade is a promising therapeutic approach to treat inflammatory diseases[45], cancer[12–14,54], and cardiovascular diseases[55]. Consistent with Caiado et al. [46], treatment with anakinra, a clinically approved IL1 receptor antagonist, reduced the myeloid expansion of *Tet2*-KO cells. In addition, we identified that *TET2*-edited human progenitors produced increased monocyte colonies upon IL1β stimulation, which is blocked by IL1 antagonism which further suggests therapeutic potential. Cumulatively, our work highlights the mechanism and therapeutic potential of inhibiting IL1β signaling for suppressing or delaying CH-associated expansion, which may apply to the more complex multifactorial CHIP-associated premalignant myeloid expansion observed within patients.

## Methods
### Transgenic animals
All mice were bred and maintained in the Department of Comparative Medicine at Oregon Health & Science University. All experiments and comparisons were performed with age and sex-balanced mice cohorts. All animal procedures were conducted in accordance with the Guidelines for the Care and Use of Laboratory Animals and were approved by the Oregon Health & Science University Institutional Animal Care and Use Committee. Mice were housed with dark and light cycles from 6am–6pm, ambient temperature at 70 degrees Fahrenheit (+/− 5 degrees), and humidity within 30%-80%. All euthanasia was performed using compressed carbon dioxide gas. A breeding colony of *Vav-Cre-Tet2*[fl/fl], described previously[6], was generously provided by Dr. Ross L. Levine from the Memorial Sloan Kettering Cancer Center, New York. Similarly, a breeding colony of *LysM-Cre-Tet2*[fl/fl] mice was generously provided by Dr. Evan Lind from Oregon Health and Science University (OHSU), Portland who acquired them from Jackson Laboratory (B6;129S-*Tet2*[tm1.1Iaai]/J, Strain # 017573). ROSA26-M2rtTA (RTA)

tetracycline-on *shTet2* mice, described previously[18], were provided by Dr. Luisa Cimmino, from New York University, New York. CD45.2[+] (C57BL/6 J, Strain # 000664), CD45.1[+] (B6.SJL-*Ptprc*[a] *Pepc*[b]/BoyJ, Strain # 0002014), *Il1r1*-KO (B6.129S7-*Il1r1*[tm1Imx]/J) mice strains were obtained from Jackson laboratory. Heterozygous CD45.1/45.2 mice were obtained by breeding CD45.1 and CD45.2 mice. Double Knockout mice (DKO) were obtained by breeding *Il1r1*-KO mice with *Vav-Cre-Tet2*[fl/fl].

### Mouse genotyping
**In house genotyping.** Tails were clipped from 3–4 weeks old mice and lysed in 100 μL buffer (2 M Tris, pH 8.8; 1 M ammonium sulfate; 0.5 M MgCl₂; 0.5% gelatin; deionized water; 2-ME; Triton X-100) containing proteinase K overnight at 55℃. DNA was diluted to 500 μL water and 2 μl was used for PCR analysis. Primers and PCRs were done as specified by protocols by Jackson Laboratory. Identification of intact *Tet2* (Protocol 19042: Standard PCR Assay - Tet2<tm1.1Iaai> Version 2.0)

*Tet2* Forward: AAG AAT TGC TAC AGG CCT GC
*Tet2* Reverse: TTC TTT AGC CCT TGC TGA GC
Identification of excision of *Tet2*
*Tet2* Forward: AAG AAT TGC TAC AGG CCT GC
*Tet2* loxp3 Reverse: TAG AGG GAG GGG GCA TAA GT
For identification of *Lys-Cre* (Protocol 26499: Separated MCA Assay - Lyz2<tm1(cre)Ifo> Version 5.2)
*Lys* mutant: CCC AGA AAT GCC AGA TTA CG
*Lys* common: CTT GGG CTG CCA GAA TTT CTC
*Lys* wild type: TTA CAG TCG GCC AGG CTG AC
Identification of *Cre* expression (Protocol 27862: Standard PCR Assay - Generic cre 2 Version 3.2)
*Cre* control Forward: CTA GGC CAC AGA ATT GAA AGA TCT
*Cre* control Reverse: GTA GGT GGA AAT TCT AGC ATC ATC C
*Cre* trans Forward: AGA TGC CAG GAC ATC AGG AAC CTG
*Cre* trans Reverse: ATC AGC CAC ACC AGA CAC AGA GAT C
*Vav-Cre-Tet2*-KO, DKO, and *Il1r1*-KO genotyping was performed by Transnetyx with the following primers:
*Tet2*[fl]
Forward Primer: CGGACGCCTCGTCAACA
Reverse Primer: GCTGGATCGGCTCTCTTATCTCATA
*Tet2*[+]
Forward Primer: GGAGGTAGCCTTGAAGAACTTGAG
Reverse Primer: GCTGGATCGGCTCTCTTATCTCATA
*Tet2*[-]
Forward Primer: CTTAAGTGTACACGCGTACTAGTCT
Reverse Primer: CAAGACCTCCCTGATGCTATGG
*iCre*
Forward Primer: TCCTGGGCATTGCCTACAAC
Reverse Primer: CTTCACTCTGATTCTGGCAATTTCG
*Il1r1*[+]
Forward Primer: CACGGCGACACCATAATTTGG
Reverse Primer: GTCCCGGTCCGCTGATATG
*Il1r1*[-]
Forward Primer: GCCTTCTTGACGAGTTCTTCTGA
Reverse Primer: CAAGAGCCATGCATGCTATTAATCC

### qPCR on *Tet2* from sorted murine BM subsets
Bone marrow was stained (staining is described in flow cytometric analysis) and -3e5 cells were sorted on the Aria III for CD45.2[+]GFP[+] and CD45.1[+] subsets. Cells were pelleted, supernatant removed, and flash frozen. RNA extraction was performed using the RNeasy Micro Kit (Qiagen, Hilden, Germany), cDNA was amplified using the High Capacity cDNA Reverse Transcription Kit (Applied Biosystems, Waltham, Massachusetts) and qPCR was run using the Power SYBR Green PCR Master Mix (Applied Biosystems, Waltham, Massachusetts) on the Quantstudio 7 Flex qPCR system. Primers were acquired from Integrated DNA Technologies (Coralville, Iowa), their sequences are described below:

*Tet2*
Forward: TGTTGTTGTCAGGGTGAGAATC
Reverse: TCTTGCTTCTGGCAAACTTACA
*Gapdh*
Forward: TCAACAGCAACTCCCACTCTTCCA
Reverse: ACCACCCTGTTGCTGTAGCCGTAT

## Bone marrow transplantation

BM transplantation was performed as described previously[14]. Briefly, bones (femurs, tibias, and hipbones) of 6 to 8 weeks old mice were flushed with DMEM containing 10% FBS to isolate marrow. Bone marrow cells were incubated with ammonium chloride solution (Stemcells Tech, Vancouver, British Columbia, Canada) for red blood cells lysis. Lineage-negative cells were magnetically enriched using a lineage cell depletion kit (Miltenyl Biotec, San Jose, California) and incubated at $37^0C$ overnight in prestimulation media (DMEM supplemented with 1% Pen/Strep, 1X L-glutamine, 15% FBS, 15% WEHI-3B, mIL-3 7 ng/mL, mIL-6 12 ng/mL, mSCF 56 ng/mL). $2.5$-$5\times10^5$ lineage-depleted cells were transplanted by retroorbital injection into lethally irradiated 6 to 8 weeks old CD45.1/CD45.2 C57BL/6 J recipient mice (in 2 doses 4 hours apart as described in the figure legends). For competition experiments, *Vav-Cre Tet2^{fl/fl}* (CD45.2) and *Lys-Cre Tet2^{fl/fl}* (CD45.2), as well as their WT control (CD45.1) donor mice, were treated with 5-fluorouracil (5-FU) 100 mg/kg retro-orbitally but not for ROSA26-M2rtTA (RTA) tetracycline-on *shTet2* mice. Non-transplant *Vav-Cre Tet2^{fl/fl}* and for ROSA26-M2rtTA (RTA) tetracycline-on *shTet2* mice competition experiments were performed without 5-FU administration. Three weeks after transplantation mice were treated with 500 ng/mouse/day of recombinant IL1β (Peprotech, Cranbury, New Jersey) intraperitoneally, with the first day of treatment constituting day 0. For the shTET2 competition experiment, doxycycline was administered orally at 1 mg/mouse. For *Vav-Cre Tet2^{fl/fl}* competition experiments treated with IL-1β and/or anakinra, anakinra was administered daily subcutaneously (100 mg/kg) 45 mins prior to the IL-1β treatments.

## Differential blood counts

Weekly, 15 µl of blood collected from the saphenous vein was mixed with equal part 2 mM EDTA and analyzed using a Vet ABC blood analyzer (Scil Animal Care Company, Barrie, Ontario, Canada) or the Element HT5 (Heska, Loveland, Colorado) according to manufacturer's directives.

## Flow cytometry analysis

BM, spleen, and PB sample collection and staining were performed as described previously[14]. Briefly, BM cells were isolated from each animal by flushing the bones with DMEM containing 10% FBS. Red blood cells were lysed using an aqueous solution of ammonium chloride (Stemcell Tech., Vancouver, Canada). All staining for flow cytometry analysis involved Fc blocking with 2.5% by volume of heat-shocked mouse serum for 10 minutes prior to staining and Fc blocking was maintained during staining. The data was collected using the Aria III cell sorter, BD LSRFortessa, and the BD Symphony A5. The data were analyzed using FlowJo software. Gating for HSPCs[22] and differentiated markers was performed as described previously[7,8].

**Weekly PB.** Weekly PB samples were stained using PE-conjugated CD45.2, PE-Cy7-conjugated CD3, APC-conjugated CD11b, and APC-Cy7-conjugated CD19 antibodies. For competition studies, cells were distinguished by CD45 isoforms using the addition of PE-CF594-conjugated CD45.1.

***Vav-cre Tet2^{fl/fl}* murine model competition experiments.** An initial cocktail combined the weekly PB cocktail described above on ~$10^5$ cells isolated from the BM, spleen, and PB samples with the addition PerCP-Cy5.5-conjugated Gr1 to help visualize terminally differentiated

populations. Additionally, APC-Cy7-conjugated B220 was used in place of APC-Cy7-conjugated CD19. A second cocktail was used on BM and spleen cells to identify HSPCs. This cocktail was comprised of FVS510-conjugated Fixable Viability Stain, PE-CF594-conjugated CD45.1, PE-conjugated CD45.2, PercP-eFluor710-conjugated CD135, Alexa 700-conjugated CD34, APC-eFluor780-conjugated cKit, PE-Cy7-conjugated CD150, Pacific Blue-conjugated Sca1, and BV605-conjugated Fc Gamma (CD16/CD32) antibodies and lineage markers (CD3, CD4, CD8, B220, Gr-1, Ter-119, CD19, IgM, and CD127 all conjugated to APC).

**For non-transplantation murine model experiments.** An initial cocktail combined the weekly PB cocktail described above with the addition of BV650-conjugated Ly-6G and BV711-conjugated Ly-6C. A second cocktail identified apoptosis and cell death within HSPCs. This cocktail was comprised of 7AAD, Annexin V, PE-conjugated Sca1, PercP-eFluor710-conjugated CD135, APC-conjugated lineage cocktail (described above), Alexa 700-conjugated CD34, APC-eFluor780-conjugated cKit, PE-Cy7-conjugated CD150, BV605-conjugated Fc Gamma (CD16/CD32) and BV711-conjugated CD48 antibodies. A third cocktail identified cell cycle by performing surface staining of PE-conjugated Sca1, APC-conjugated lineage cocktail (described above), Alexa 700-conjugated CD34, APC-eFluor780-conjugated cKit and BV605-conjugated Fc Gamma (CD16/CD32) for 30 minutes followed by fixation and permeabilization using BD Cytofix/Cytoperm (BD Biosciences, San Jose, California). Intracellular staining using FITC-conjugated KI-67 was performed overnight at 4 degrees, followed by staining with DAPI at with a final concentration of 0.5ug/ml.

**ROSA shTET2 transplantation experiments.** The same staining cocktails were used as the non-transplantation model, with the following exceptions. GFP co-expressed with shRNA targeting *Tet2* was analyzed. PE-conjugated CD45.2 and PE-CF594-conjugated CD45.1 delineated CD45 isoforms. Pacific Blue-conjugated Sca1 was used in place of PE-conjugated Sca1.

**CD45.1 versus CD45.2 transplantation experiments.** The transplantation was performed as described above and PB was stained with FITC-conjugated CD45.2 and PE-ef610-conjugated CD45.1.

## Murine colony formation assay

BM cells were harvested from wild-type and *Tet2*-KO mice (N = 4, 8–12 weeks old). Cells were subjected to red cell lysis in 5 ml ammonium chloride lysis buffer solution (Stem Cell Technologies) on ice for 10 minutes. Lineage cells were depleted as described above. Cells were kept overnight in DMEM media containing 15% fetal bovine serum, 1% pen-strep, 2% L-Glutamine, 15% WEHI, mIL-3 (7 ng/mL), mIL-6 (12 ng/mL), and mSCF (56 ng/mL). Cells were stained with PE-conjugated Sca1, APC-conjugated lineage cocktail, APC-eFluor780-conjugated cKit and 7AAD and purified by fluorescent activated cell sorting (FACS) at a density of 600 Lin⁻Sca1⁺cKit⁺7AAD⁻ cells per 1.1 ml/well of MethoCult™ M3334 (STEMCELL Technologies, Vancouver, British Columbia, Canada). M3334 methocult was supplemented with cytokines to a final concentration of 5 ng/ml mIL-3, 25 ng/mL mSCF, and with or without 25 ng/ml mIL1β. Colonies were counted one week after plating and categorized as colony formation units (CFU); CFU-G, CFU-M, CFU-GM or CFU-E. Cells from the colony formation assay were then resuspended in IMDM 2% FBS and re-plated at $1.2 \times 10^5$ cells/well in identical conditions up to the quaternary plating.

## CRISPR-editing for *TET2* using human CD34⁺ cells and colony formation assay

Healthy donor bone marrow was purchased commercially (Lonza inc). CD34⁺ progenitor cells were enriched using magnetic CD34 microbead kit (Miltenyl Biotec, San Jose, California) and expanded overnight in SFEMII (Stem Cell Technologies) supplemented with SCF (100 ng/mL),

TPO (100 ng/mL), FLT3-ligand (100 ng/mL), IL-6 (100 ng/mL), LDL (10 ug/mL), and UM171 (38 nM) (Peprotech, Cranbury, New Jersey). Cells were resuspended in Amaxa P3 Primary Cell solutions, and added to RNP complexes consisting of either a non-targeting scrambled sgRNA (Negative Control sgRNA (mod) #1; Synthego) or the multi-guide sgRNA (G1: AGAGCUCAUCCAGAAGUAAA; G2: UUAUGGAAUACCCU-GUAUGA; G3: UCCUCCAUUUUGCAAACACU) provided with the Gene Knockout Kit v2 for human TET2 (Synthego, Redwood City, California). Cells were nucleofected with the program DZ-100 with a Lonza 4-D Nucleofector, and then immediately transferred to SFEMII. 24 hrs after nucleofection editing efficiency was quantified and we achieved an initial indel frequency of ~16%. For this PCR was performed following primers: 5′-TCAACTAGAGGGCAGCCTTG-3′ (F) and 5′-TGGCTTACCCCGAAGTTACG-3′ (R), and conditions: 97 °C for 5 min, then 35 cycles of 92 °C for 30 s, 60 °C for 30 s, 72 °C for 20 s, and a final extension at 72 °C for 10 minutes. Sanger sequencing of PCR product was performed using forward primer and subsequently analyzed using Synthego ICE analysis tool. 24 hrs after nucleofection, cells were also plated at 600 cells per well in H4434 methocult with or without 25 ng/ml hIL1β. Colonies were counted two weeks after plating and categorized as colony-forming units (CFU); CFU-G, CFU-M, CFU-GM or CFU-E. Individual colonies were picked, PCR amplified and Sanger sequenced to determine indel frequency as described above.

## Single cell RNA sequencing

BM cells were isolated from *Vav-Cre Tet2^fl/fl Tet2* KO and *Tet2^fl/fl* (WT) mice treated with IL1β or vehicle for 5 weeks (N = 4 per each genotype/condition, 8-12 weeks old) and lineage depleted using direct lineage cell depletion kit (Miltenyl Biotec, San Jose, California). This step allows the enrichment of lineage-negative cells by ~5 fold while maintaining a representation of mature populations but at a lesser extent. The cells from each mouse were stained with TotalSeq™ (Biolegend) cell hashing antibodies (Hash Tag 1: ACCCACCAGTAAGAC and Hash Tag 2: GGTCGAGAGCATTC) and washed. Two samples were mixed in equal proportions from two mice of the same genotype and treatment type, 8 sample pools total (16 samples). Each sample pool was loaded onto Chromium Controller (10x Genomics) to form single cell gel bead emulsions (GEM) and reverse transcribed generating barcoded cDNA. Transcriptome (RNA) and hashtag (HTO) libraries were size selected using SPRI-beads and sequenced separately on a NovaSeq-6000 instrument.

## Single cell RNA-seq data analysis

A total of 54,984 barcoded single-cell libraries were generated from the 10x scRNA-seq experiment. Cell Ranger package (10x Genomics) was used to align mRNA reads from 10x scRNA-seq libraries to murine GRCm38 (mm10) genome and generate feature-barcode matrices.

Downstream analysis was performed using Seurat (v3) package with a set of standard vignettes[56]. Cells with n feature counts <750 and mitochondrial DNA content >10% were removed as low-quality/apoptotic cells. HTO libraries from hashtag antibody cell labeling were used to assign each cell to individual mice. Cells labeled with two HTO barcodes were excluded as potential doublets.

Dataset integration was performed using the reciprocal PCA method with *Tet2* WT (vehicle-treated) as a reference. Dimensionality reduction and clustering revealed 22 transcriptional clusters. Cell type assignment to each cluster was performed based on top marker expression per cluster and comparison to haemopedia RNA-seq reference[57]. These were then further cross-checked against known genes associated with specific lineages and cell types leading to 17 clusters[32,58,59]. Differentially expressed genes were identified between *Tet2* KO and WT as well as a vehicle vs. IL1β treatment for each cell type using 'FindMarkers' function in Seurat. DESeq2 analysis was performed to verify significant genes considering replicate samples.

For pseudotime analysis, we followed the standard analysis vignette in the Monocle3 package[60]. Gene module signature and percentage of cells per conditions over pseudotime were constructed using custom R functions, code available on our Github page.

## Differential gene expression analysis of AML TET2 variants samples

Normalized gene expression data produced by bulk RNA-seq for AML samples were obtained from https://biodev.github.io/BeatAML2/. Sample information and data processing pipelines were described previously[61,62]. Expression data was filtered for primary tumors only using sample metadata fields 'isRelapse' = 'FALSE' and 'isDenovo' = 'TRUE' and split into TET2 and non-TET2 variant-containing groups. Differential gene expression analysis between these two groups was performed using the 'limma' package[63] and assessed using empirical Bayes moderated t-test[64]. p-Values were corrected based on the Benjamini-Hochberg approach[65].

## DNA methylation library preparation

Whole genome bisulfite sequencing (WGBS) libraries were prepared from FACS purified populations of LSK, CMP, and GMP cells using an ultra-sensitive method developed for single-cell sequencing scBS-seq[66] and G&T-seq[67]. Briefly, ~500 flow-sorted cells were resuspended in 2.5 μl of RLT plus buffer (Qiagen). Polyadenylated RNA was captured with oligo-dT conjugated beads and processed using Smart-seq2 protocol[68]. The remaining genomic DNA was bisulfite converted (EZ DNA methylation kit, Zymo) and the protocol for single-cell BS-seq library preparation was followed[66]. A total of 48 libraries were prepared (N = 4 animals per genotype/treatment, 3 cell types, *Tet2* WT/KO + /-IL1β), pooled using appropriate Illumina Truseq CD i5/i7 index adapters, and sequenced on a Novaseq-6000 instrument (Illumina) to the depth of ~20x for LSK and ~4x for CMP and GMP samples.

## DNA methylation analysis

Bisulfite sequencing (BS-seq) libraries were demultiplexed by the sequencing facility and processed on a slurm compute cluster aligning to GRCm38 murine genome. A detailed description of the alignment pipeline is described elsewhere[69]. Briefly, adapter trimming was performed using trim_galore with following options: trim_galore –gzip –clip_r1 6 –clip_r2 6 –paired –fastqc

Genomic alignment were performed using bismark[70] aligner in a non-strand specific manner separately for read one and two:

bismark --non_directional --gzip -N 1 --parallel 4 --unmapped --score_min L,−10,−0.2

DNA methylation calls were extracted from the aligned reads as CpG coverage files and analyzed using the SeqMonk application. Differentially Methylated regions were defined in SeqMonk using windowed probes tiling 50 CpGs sites across the mouse genome. Probes with ≥10 CpGs with at least one read were quantified for average DNA methylation level using a SeqMonk in-built bisulfite sequencing quantitation pipeline. A logistic regression model was applied to define DMRs between *Tet2* WT vs KO cells with both vehicle or IL1β treatment conditions. Probes with p-value < 0.05 and absolute DNA methylation difference ≥10% were called as biologically significant.

Enhancer elements for different blood lineages were defined based on publicly available reference histone modification datasets[71]. Briefly, raw data was downloaded from the GEO database (GSE59636) and re-aligned to GRCm38 (mm10) murine genome. Enhancers for HSC, CMP, GMP, and MEP cells were defined as follows: first, H3K27ac peaks were called using Model-based analysis of ChIP-seq (MACS)[72] within the SeqMonk analysis tool. Second, H3K27ac peaks that did not overlap known promoter elements and had a higher H3K4me1/H3K4me3 ratio were called as enhancers for each cell type. The CpG methylation of probes overlapping with an enhancer site for one and

only one cell state (MEPs, CMPs, GMPs, CLPs) were defined as specific probes and the rest were excluded.

DNA binding proteins constituting transcription factors and epigenetic regulators with known roles in hematopoiesis were chosen and supplemented with HOMER-identified transcription factors with literature-derived significance. These were limited to those with publicly available ChIP-seq binding data within embryonic stem cells or hematopoietic lineages (44 proteins). ChIP-seq "BED" files of called peaks were downloaded from Cistrome. When necessary, MGSCv37 (mm9) genome alignment was lifted to GRCm38 via UCSC "LiftOver". SeqMonk performed quantitation of CpG methylation across defined probes spanning all DNA binding protein called peaks (784,110 probes). Of the 44 proteins chosen for ChIP-seq, 8 were excluded because they lacked sufficient probes with coverage (>100 probes) for at least one of the 48 samples, leaving 36 remaining factors (Supplementary Data 6). The difference in CpG methylation between each individual probe between *Tet2*-KO and WT was calculated. Values were referenced against the average difference between all CpGs with sufficient coverage for LSKs, CMPs and GMPs to identify transcription factors that possessed unusually high methylation.

To identify specific factors binding sites which may exhibit altered transcription due to methylation, ChIP binding sites of the 36 remaining factors were annotated by SeqMonk to determine genes within 2k base pairs with significant (*p* < 0.05) methylation changes. These genes were compared with the top 100 most significantly downregulated genes in *Tet2*-KO relative to WT. These genes were then manually annotated for function by literature search.

### Gene set enrichment analysis

Gene set enrichment analysis pre-ranked was performed using the GSEA software v4.2.1 from the Broad Institute. Genes in *Tet2*-KO vs WT cells with or without IL1β were ranked by the sign of the log fold change in gene expression multiplied by the negative log base ten of the p-value. Analysis was performed for murine datasets using the ChIP platform Mouse_Gene_Symbol_Remapping_MSigDB.v7.0. Molecular signature databases include the Broad Institutes' "C2.all.v7.4symbols", as well as the literature-derived genes identified as markers of the aged specific LT-HSCs cluster[25]. Analysis of DEGs from AML samples using Human_Gene_Symbol_with_Remapping_MSigDB.v7.5.1.chip. Molecular data bases were derived from the top 100 genes DEGs between *Tet2*-KO relative WT (Supplementary Data 2) which were converted to human analogues using Ensembl BioMart.

### Enrichr and STRING analyzes

For transcriptional data, statistically significant genes below a (q < 0.05) which were upregulated in *Tet2*-KO versus WT HSCs in either vehicle or IL1β treated mice were used for Enrichr analysis using "KEGG 2021 Human" dataset. For methylation data genes within 5000 base pairs of differentially methylated genes were utilized. STRING analysis (v11.5) was performed using genes significantly upregulated (q < 0.05) in *Tet2*-KO versus WT in both vehicle or IL1β treated mice. Data is from "Multiple proteins" analysis using *mus musculus* data and unconnected nodes by the full STRING network were removed. Gene ontology and KEGG pathway analysis was used from STRING analysis.

### HOMER analysis

Differentially methylated regions of interest spanning 50 CpG length tiles described above in this manuscript with a statistically significant increase or decrease of at least 10% in methylation level in *Tet2*-KO relative to WT were used as input for Homer's findMotifsGenome.pl script along with the mouse reference genome GRCm38. DMRs containing motifs of interest were determined by using Homer's annotatePeaks.pl script with the genes/DMRs and motif results.

### Statistics & reproducibility

To determine significance, a two-factor or three-factor ANOVA was performed using the Graphpad Prism software and FWER adjusted p values are reported. Alternatively, where appropriate a comparison was performed using a pairwise Student's two tailed t-test using the Microsoft excel software: (*): *p* < 0.05; (**): *p* < 0.01; (***): *p* < 0.001; (****): *p* < 0.0001. For single cell RNA sequencing *p* values were adjusted Benjamini-Hochberg through DESeq2 to produce false discovery rate adjusted q values. Similarly, bisulfite sequencing by 50 CpG tiling for differentiation methylation by SeqMonk, as well as HOMER, STRING, and Enrichr were each adjusted for false discovery rate (FDR) by their respective software. All figure legends specify the statistical method used to identify the significance of each panel. Orthogonal experimental model systems and approaches were taken to ensure the reproducibility of findings. Sample size estimation aimed for as close to sample sizes of 6 as possible to detect with 90% confidence up to a 50% change in the magnitude of difference in phenotypes with alterations due to the availability of mice genotypes at the initiation of each experiment. The Investigators were not blinded to allocation during experiments and outcome assessment.

### Reporting summary

Further information on research design is available in the Nature Portfolio Reporting Summary linked to this article.

## Data availability

The single RNA sequencing and bisulfite sequencing data generated in the study has been deposited in the Gene Expression Omnibus (GEO) under accession number GSE210026. Source Data for all mouse experiments have been provided. All other data are provided in the Supplementary Information/Source Data file. Source data are provided with this paper.

## Code availability

The analyzes presented in this manuscript did not involve any custom code or custom statistical algorithms and did not involve any modifications to the libraries/packages used. Further details and code are available from the corresponding authors upon request. All code is available on our Github page. https://github.com/rusja-s/Tet2_inflammation_ms_code#tet2_inflammation_ms_code.

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

## Acknowledgements

We thank OHSU Massively Parallel Sequencing Shared Resource, Flow Cytometry Core, and OHSU Department of comparative medicine for their support. We also thank Joseph Estabrook for providing feedback on the single-cell analysis approach. We further thank Andy Kaempf for providing biostatistical expertize. Funding for this project was provided by the American Cancer Society (RSG-17-187-01-LIB) as a research scholar award (AA) as well as Knight Cancer Research Institute and Cancer early detection advanced research center (Oregon Health & Science University) as pilot research projects. AA is also supported by grants from the National Cancer Institute (R01 CA229875-01A1), National Heart, Lung, and Blood Institute (R01 HL155426-01), Alex Lemonade /Babich RUNX1 Foundation, EvansMDS Foundation, and V Foundation Scholar award (AA). JM is supported by a Cancer Biology Student stipend grant and the National Heart, Lung, and Blood Institute NRSA F31.

## Author contributions

A.A. provided project oversight for experimental design, data analysis, interpretation, and methods development; J.M., H.L., M.M., E.W., C.E., D.G., and A.A. performed all the experiments and analyzed the data. R.S., J.M., A.A., B.A.D., and H.M. performed single cell and whole-genome bisulfite sequencing experiments and analysis. P.C. and G.W. analyzed the BeatAML dataset. L.C. and W.H.F. provided critical resources and guidance for in vivo experiments. All of the authors wrote the manuscripts and provided feedback.

## Competing interests

The authors declare no competing interests.

## Additional information

[1]Division of Oncological Sciences, Oregon Health & Science University, Portland, OR, USA. [2]Department of Cell, Developmental, and Cancer Biology, Oregon Health & Science University, Portland, OR, USA. [3]Cancer Early Detection Advanced Research Center, Knight Cancer Institute, Oregon Health & Science University, Portland, OR, USA. [4]Division of Hematology & Medical Oncology, Oregon Health & Science University, Portland, OR, USA. [5]Division of Pediatric Hematology and Oncology, Oregon Health & Science University, Portland, OR, USA. [6]Department of Medical Informatics and Clinical Epidemiology, Oregon Health & Science University, Portland, OR, USA. [7]University of Miami, Department of Biochemistry and Molecular Biology, Sylvester Comprehensive Cancer Center, Miami, USA. [8]Department of Molecular and Medical Genetics, Oregon Health & Science University, Portland, OR, USA. [9]These authors contributed equally: J. McClatchy, R. Strogantsev. [10]These authors jointly supervised this work: H. Mohammed, A. Agarwal . ✉e-mail: agarwala@ohsu.edu

