## [Peer Review File · Nature Communications]

Clonal-hematopoiesis-related TET2 loss-of-function impedes IL1 β -mediated epigenetic reprogramming in hematopoietic stem and progenitor cellsREVIEWER COMMENTS

Reviewer #1 (Remarks to the Author):

In this paper, McClatchy J et al. are assessing the role of inflammation driven by IL1B in the clonal expansion and impact on hematopoiesis of Tet2 deficient mice as a model for human age-related clonal hematopoiesis (ARCH, but in reality CHIP- see below), which has been recently recognized as a premalignant lesion , and a risk factor for other disorders such as cardiovascular diseases.

They used several mice models and molecular strategies to analyzed the impact of chronic stimulation with IL1B on Tet2-KO cells. They subjected Tet2-KO mice transplanted with equal numbers of Tet2 deficient cells and WT cells to chronic IL1B exposure. This favored Tet2-KO cell proliferation and induced a myeloid proliferation bias. Notably, IL1B1 stimulation also favored a myeloid bias in normal cells but to a lesser extent. IL1B also reduced T-cell frequency in Tet2-KO cells in comparison to WT ones. They also showed that chronic IL1B exposure favored the expansion of LT-HSC in contrast to WT cells. Supporting expansion and differentiation.

They also demonstrated that that Tet2 KO cells under the influence of IL1B increase self-renewal in contrast to WT cells. Their observations are supported by a series of single cell expression analyses that demonstrate enrichment of transcriptional signatures associated with chronic stress, self-renewal and myeloid priming.

They performed WGBS and demonstrated higher methylation status of Tet2-KO cells which were increased upon IL1B stimulation.

Then they tested, if abrogating of the IL1B-receptor would affect their results. They showed that IL1B-receptor deficient mice abrogated in part the phenotype induced by IL1B exposure, re-enforcing its role as a driver of the phenotype.

Interestingly, they used the commercially approved IL1R antagonist Anakinra to see if it would impede the phenotypes observed in their models. They show that it reduces the myeloid proliferation bias of Tet2-KO cells.

They conclude IL1 β enhanced the self-renewal ability of Tet2-KO HSPCs by upregulating genes associated with self-renewal and by resisting the demethylation of binding sites of transcription factors promoting terminal differentiation. Using aged mouse models and human progenitors, they hypothesized that targeting IL1 signaling may be an early efficient strategy in preventing clonal progression in human with CHIP.

General Comments.

This is a very timely paper as the medical community has just recognized for the first time the premalignant nature of Clonal Hematopoiesis (WHO 2022). However, elaboration of early intervention and prevention strategies needs to be based on the understanding of the biologic features and risk factors associated with clonal expansion and progression. This well conducted series of experiment further consolidate the role of chronic inflammation in clonal expansion of CH. It may, on its own, further support clinical intervention in high-risk patients. However, Human CH is far more complex than the model used in the described studies. There is still controversy about the exact role of inflammation in CH in human. Further, what may apply for certain mutations and may be different for others drivers.

Suggestions for revision (I will leave comment on experimental mice studies to reviewers' expert on this field).

1. Title: it is not TET2 that cooperates, it is its lack of function: reformulate.
2. There is a bit of confusion in the definition of CH vs ARCH and CHIP in the intro. please refer to the definition of WHO 2022. I suggest to use CHIP for Human and CH for the animal models.
3. There are different studies that clearly indicated that the inflammatory driver of myeloid proliferation was IL6 (cited by the authors), It would be interesting to elaborate on this in the discussion. Is there alternate pathways, eventually suppressing one or the other will also lead to

suppressing bad clones?

4. As stated earlier, it is important to specify in the discussion that CHIP in human is more complex and that many other factors other than inflammation may play a role. (This not precluding clinical trial in appropriate subjects with anti-inflammation intervention, there is one that has already started in CCUS using anakinra).

Reviewer #2 (Remarks to the Author):

General comment

In their study, McClatchy J. et al. address the molecular mechanisms underlying IL-1B-mediated Tet2-KO clonal hematopoietic expansion. The research question builds on recent studies, highlighting inflammation and particularly IL-1 as a driver of Tet2-mutant leukemogenesis. In particular, a recent screen in up to 22092 individuals with clonal hematopoiesis revealed that increased IL-1B circulating levels associate significantly and exclusively with TET2 mutant clonal hematopoiesis (Bick AG et al. Nature 2020). Moreover, mouse studies have implicated increased IL-1 levels in aged animals as a driver of Tet2-mutant clonal expansion (Caiado F, et al. Blood 2022; Burns SS, et al. Leukemia 2022). While the observation that IL-1 drives Tet2-mutant clonal expansion is already well established and not novel, the precise molecular underpinnings are not addressed yet, and they are of key relevance to the field.

To address this, the authors used different bone marrow (BM) chimera setups carrying WT and Tet2KO hematopoiesis and observed that prolonged IL-1B exposure leads to expansion of the Tet2KO myeloid compartment (particularly CD11b+Gr1^{high} subset) in BM, blood and spleen (Sp). Additionally, IL-1 exposure led to a general but heterogeneous expansion of the BM and Sp HSPC compartment (MPP2 in BM and LT-HSCs/MPP2-4/CMPs/GMPs/MEPs in Sp). Functional analysis of WT or Tet2KO LSKs exposed to vehicle or IL-1 revealed increased re-plating capacity (an in vitro measurement of self-renewal) of IL-1 Tet2KO LSKs. To identify the molecular mechanisms underlying this IL-1 mediated expansion, the authors performed sc-RNA seq on lineage-depleted BM cells from WT and Tet2KO mice treated with vehicle or IL-1 for 5 weeks. Sc-RNA seq data revealed increased expansion of IL-1 treated Tet2KO granulocyte/monocyte progenitor cluster. Focusing on the HSCs cluster, the authors identify increased transcriptional signatures associated with chronic stress, self-renewal and myeloid priming, and decreased signatures associated with proliferation and lymphopoiesis in IL-1 treated Tet2KO samples. Whole genome bisulfite sequencing in purified LSKs, CMPs and GMs from vehicle or IL-1 treated Tet2KO or WT mice revealed that Tet2KO resist IL-1-driven methylation reduction of lineage-specific enhancer regions and terminal differentiation promoting transcription factor binding sites. This data suggests that failure of Tet2KO cells to demethylate these sites upon IL-1 exposure may account for their enhanced self-renewal capacity. Having established that IL-1 drives Tet2KO clonal expansion and myeloid bias, the authors performed genetic ablation or pharmacological inhibition (using IL1R1 antagonist anakinra) of Il1r1 in WT or Tet2 KO mice and observed a delayed myeloid and progenitor expansion, splenomegaly and overall survival of Tet2;Il1r1KO mice. To validate the findings in human cells, the authors deleted Tet2 from human progenitors and confirmed their IL-1 induced expansion in vitro.

Overall, this is a relevant and generally well-performed study, confirming previous findings on IL-1 as a targetable driver of Tet2-mutant clonal expansion and leukemogenesis. The novelty of the study lies in describing the underlying molecular aspects (transcriptional and epigenetic) by which IL-1 drives Tet2-KO myeloid expansion.

Specific comment

Major comments (in random order)

1. The BM chimera approach used in figure 1 has multiple limitations that need to be addressed:
a) The treatment with IL-1 starting at 3 weeks post transplantation is likely somewhat too early,

since multilineage repopulation after total BM transplantation takes at least 8-12 weeks for complete reconstitution. This fact is evident when considering the frequency of PB CD11b+ cells in untreated WT CD45.1+ fraction, which is 90% at week 3 and only stabilizes to the expected 15% (PMID: 18544662) by week 10. The experiment should be performed starting the IL-1B treatment the earliest by week 8 post transplantation to avoid confounding inflammatory signals derived from irradiation.

b) The fact that WT and Tet2KO are in different CD45 isoforms might explain in part the excessive expansion of Tet2KO (even in the absence of IL-1) and might confound the effect of IL-1B. Indeed, a 25% reduction in homing efficiency, 3.8-fold reduction in transplantable long-term hematopoietic stem cells, a 5-fold reduction in LT-HSCs capable of 24-hour homing, and a cell-intrinsic engraftment defect of 30% to 50% was reported in B6.SJL-derived bone marrow cells (CD45.1) relative to B6-derived cells+ total BM (CD45.2) (PMID: 19901263). The authors should add two control groups with WT CD45.2 competing with WT CD45.1 in the presence of vehicle or IL-1B.

2. A key aspect of the study is the expansion of Tet2KO HSPCs upon IL-1B treatment. However, the authors do not show this in the competitive chimera setup and rather choose to show this in the more complex experimental setup using the doxycycline-inducible shRNA targeting Tet2. Given the importance of this observation, they should show the BM HSPC chimerism relative to figure 1a. Moreover there is some conflicting data concerning the expansion of the HSPC compartment when using FACS (figure 2) and scRNA-seq. This discrepancy needs to be clarified.

Minor comments

1. Please clarify the meaning of the IL-1B arrows in figure 1e.
2. Tet2 expression in GFP+ and - cells should be shown (relative to Figure 1d-e).
3. In SuppFigure 2B the initial levels of CD11b+ cells seem too high (50% when in general it should be around 15-20%). Can the authors comment on this?
4. Group comparisons statistical analysis are done using independent t-testing. However, for multiple comparisons the authors should preform A-NOVA. This would make the significant differences more clear.
5. When describing figure 2A the authors claim increased LT-HSC frequency in shTet2 upon IL-1B stimulation compared to vehicle. However, this increase is not significant. This needs to be clearly indicated.
6. In cell cycle analysis, the authors should separate G0 from G1 fractions.
7. Figure numbering needs to be carefully checked (e.g. page 9, line 200 I believe the authors are referring to figure 2d).
8. Frequency of BM HSPCs populations (LT-HSCs, MPPs...) concerning anakinra treatment (relative to Figure 5e-g) should be shown.
9. Recent data by Caiado et al. (PMID: 36379023) shows that IL-1 is a driver of Tet2-mutant clonal hematopoiesis. This data is highly relevant for the current study and should be cited and discussed in context of the here presented findings.

Reviewer #3 (Remarks to the Author):

The manuscript by McClatchy et al presents results demonstrating the relationship between IL1b and loss-of-function TET2 mutations in promoting an increased self-renewal ability of the hematopoietic stem and progenitor cells (HSPC), a myeloid-skewed hematopoiesis, and the differentiation of pro-inflammatory MHCII+ macrophages. To do so, the authors used a transplantation model using HSPCs from the Tet2-KO mouse strain developed by Aifantis I and Irvine RL (reference 6), and investigated hematopoiesis and hematopoietic progenitors upon IL1b stimulation. They also used an inducible-shTET2 mouse model to mimic the acquisition of somatic mutations in Tet2 (reference 19). The study was well conducted and results supported the conclusions of the authors. However, some issues remain to be clarified.

Major comments:

- The authors deeply described the bias observed in the mature neutrophils and monocytes/macrophages subsets in spleen and bone marrow (BM) upon stimulation by IL1b. In

the Tet2-KO mouse model, T-cells were reduced in Tet2-KO mice compared to WT with or without addition of IL1b whereas B-cells were reduced only in presence of IL1b. What are the results concerning lymphocytes percentages and counts in the inducible mouse model? Moreover, the Tet2-KO mice showed increased percentages of lymphocytes in blood compared to WT in supplemental figure 2c. These results look contradictory with the ones showed in supplemental figure 1b and would need to be commented.

- In figure 3g, the authors used the scRNAseq signature based on the differentially expressed genes found in Tet2-KO vs WT HSCs at steady and pro-inflammatory states to analyze TET2 mutated AML primary samples. The choice to study primary AML is rather surprising as TET2 is not the most frequently mutated gene in this disease by contrast to myelodysplastic syndromes (MDS) (Ogawa S. Blood 2019). Moreover, because of the continuum in the progression from clonal hematopoiesis to MDS and then to secondary AML, MDS seem to be more relevant as it better corresponds to the model of study developed by the authors. Applying the signature to TET2-mutated MDS samples compared to TET2 WT would probably provide interesting results.

- The analysis of the DNA methylation in the Tet2-KO LSK compared to WT upon IL1b treatment showed hypermethylation of the enhancers, notably for lymphoid genes. The authors performed this experiment on LSKs, CMPs and GMPs. However, what is the DNA methylation landscape in the Common Lymphoid Progenitor (CLP) subset in presence or absence of IL1b?

- A recent report showed that IL1a and IL1b do not play redundant roles during immune response (Eislmayr et al Science 2022). Even more recently, an article by Caiado et al suggested a major role for IL1a in the age-driven Tet2+/- clonal hematopoiesis (Caiado et al Blood 2023). How do the authors reconcile their observations with the one by Caiado et al? Does IL1a collaborate with IL1b in their model?

Minor comment:

- Pages 9-10. There is no panel f, nor g, in figure 2.

Nicolas DULPHY.

Reviewer #4 (Remarks to the Author):

McClatchy and colleagues investigated the effect of IL1 β signaling on HSPCs in a model system of clonal hematopoiesis (CH). To study the development of hematological malignancies in CH they used a murine TET2 loss-of-function model, as TET2 is one of the most common mutations in CH. The authors explored the changes in cell type abundancies upon TET2 KO and IL1 β stimulation using flow cytometry and could recapitulate the known expansion of HSCs and the myeloid bias of HSPCs in response to pro-inflammatory stress. By using single-cell RNA sequencing of 54,984 lineage-depleted bone marrow cells, as well as whole genome bisulfite sequencing of selected hematopoietic subpopulations, the authors supported their initial findings and further elucidated the mechanism of the observed myeloid bias. They could show that TET2 KO LSK cells had higher global methylation than WT cells and this methylation was enriched in lymphoid enhancer sites. Finally, the authors could reverse the TET2 KO and IL1 β induced phenotype by using a clinically approved IL1 receptor antagonist, thus providing a promising therapeutic approach. The authors made a great effort to validate their findings in several different mouse models (TET2 KO mice, a doxycycline-inducible shRNA targeting TET2) and with different techniques (Chimerism experiments, scRNA seq, WGBS). However, it is unclear how novel the findings are and how many more insights beyond existing literature (for example Caiado et al. Blood 2023) the authors gained.

Major comments:

1. As stated above, besides the methylation analysis it is not clear to the reader which findings are new and valuable to the field and which findings confirm existing findings. A critical discussion of what makes this study outstanding with respect to other findings is needed.

2. For the scRNA seq part, the authors showed gene expression of marker genes for individual cells from each cluster, the frequency of cell types per experimental condition, and a combined UMAP. However, metadata such as the number of sequenced cells per condition and animal is missing. In addition, the analysis of the scRNA seq is not very elaborate and focuses mainly on DEG analysis in HSCs. It would strengthen the point of the paper if the author would also include an analysis of subpopulation as they did in figure 4. Other analysis such as trajectory analysis of the single conditions could be helpful to visualize the observed myeloid bias.

Minor comments

3. Supplemental figures regarding the sorting strategy of the flow cytometry analysis are missing, especially for figure 1, which could be helpful to interpret the depicted results.

4. Furthermore, the graphical illustration of their results is often crowded (such as Figure 2c, 5c) and needs revision.

5. The authors used a pairwise Student's two tailed t-test for analyzing most of their experiments, but often they compare more than two variables. It is not clear to the reader if the p-value was corrected for multiple testing.

IL1 β and TET2 cooperate to drive lineage-dependent transcriptional and epigenetic reprogramming in clonal hematopoiesis.

We thank the Editor and the reviewers for their valuable, thoughtful, and constructive comments that have significantly improved our study. Please find outlined below our detailed response to each point raised by the reviewers. The corresponding changes in the manuscript are highlighted in Blue. Additionally, we have included a few critical figures in our response letter for the convenience of the reviewers.

Reviewer # 1's comments:

In this paper, McClatchy J et al. are assessing the role of inflammation driven by IL1B in the clonal expansion and impact on hematopoiesis of Tet2 deficient mice as a model for human age-related clonal hematopoiesis (ARCH, but in reality CHIP- see below), which has been recently recognized as a premalignant lesion, and a risk factor for other disorders such as cardiovascular diseases. They used several mice models and molecular strategies to analyzed the impact of chronic stimulation with IL1B on Tet2-KO cells. They subjected Tet2-KO mice transplanted with equal numbers of Tet2 deficient cells and WT cells to chronic ILB1 exposure. This favored Tet2-KO cell proliferation and induced a myeloid proliferation bias. Notably, IL1B1 stimulation also favored a myeloid bias in normal cells but to a lesser extent. IL1B also reduced T-cell frequency in Tet2-KO cells in comparison to WT ones. They also showed that chronic IL1B exposure favored the expansion of LT-HSC in contrast to WT cells. Supporting expansion and differentiation. They also demonstrated that that Tet2 KO cells under the influence of IL1B increase self-renewal in contrast to WT cells. Their observations are supported by a series of single cell expression analyses that demonstrate enrichment of transcriptional signatures associated with chronic stress, self-renewal and myeloid priming. They performed WGBS and demonstrated higher methylation status of Tet2-KO cells which were increased upon IL1B stimulation. Then they tested, if abrogating of the IL1B-receptor would affect their results. They showed that ILB1-recepror deficient mice abrogated in part the phenotype induced by IL1B exposure, re-enforcing its role as a driver of the phenotype. Interestingly, the use the commercially approved IL1R antagonist Anakinra to see if it would impede the phenotypes observed in their models. They show that it reduces the myeloid proliferation bias of Tet2-KO cells. They conclude IL1 β enhanced the self-renewal ability of Tet2-KO HSPCs by upregulating genes associated with self-renewal and by resisting the demethylation of binding sites of transcription factors promoting terminal differentiation. Using aged mouse models and human progenitors, they hypothesized that targeting IL1 signaling may be an early efficient strategy in preventing clonal progression in human with CHIP.

General Comments.

This is a very timely paper as the medical community has just recognized for the first time the premalignant nature of Clonal Hematopoiesis (WHO 2022). However, elaboration of early intervention and prevention strategies needs to be based on the understanding of the biologic features and risk factors associated with clonal expansion and progression. This well conducted series of experiment further consolidate the role of chronic inflammation in clonal expansion of CH. It may, on its own, further support clinical intervention in high-risk patients. However, Human CH is far more complex than the model used in the described studies. There is still controversy about the exact role of inflammation in CH in human. Further, what may apply for certain mutations and may be different for others drivers.

Response: Thank you for highlighting that our study is timely and has high clinical significance.

Suggestions for revision (I will leave comments on experimental mice studies to reviewers' expert on this field).

1. Title: it is not TET2 that cooperates, it is its lack of function: reformulate.

Response: We agree that this is not cooperation but loss of function, and have edited the title from: "IL1 β and TET2 cooperate to drive lineage-dependent transcriptional and epigenetic reprogramming in clonal hematopoiesis."

To:

"IL1 β in the context of TET2 loss-of-function drives lineage-dependent transcriptional and epigenetic reprogramming in clonal hematopoiesis."

2. There is a bit of confusion in the definition of CH vs ARCH and CHIP in the intro. please refer to the definition of WHO 2022. I suggest to use CHIP for Human and CH for the animal models.

Response: The "The 5th edition of the World Health Organization Classification of Haematolymphoid Tumours: Myeloid and Histiocytic/Dendritic Neoplasms" has been used to clarify any terminology related to the use of CHIP and CH within the manuscript. In agreement with Reviewer 1's suggestion, CHIP now refers exclusively to human models harboring somatic mutations of myeloid malignancy-associated genes detected in the blood or BM at a VAF greater than or equal to 2% without other hematological diagnosis. To distinguish itself from this more clinical definition, CH is used to refer broadly to the expansion of a mutant clonal population bearing selective growth advantage within our murine models' experiments.

3. There are different studies that clearly indicated that the inflammatory driver of myeloid proliferation was IL6 (cited by the authors), It would be interesting to elaborate on this in the discussion. Is there alternate pathways, eventually suppressing one or the other will also lead to suppressing bad clones?

Response: This is an excellent suggestion; we have included in the discussion on page 19 regarding the role of IL-6 and implications of alternate cytokines signaling pathways as potential drivers via different mechanisms. This text is found below:

“We found no significant differences in apoptosis between *Tet2*-KO and WT HSPCs upon IL1 β stimulation, whereas reductions in apoptosis of *Tet2*-KO HSPCs have been shown upon chronic exposure to TNFA or acute exposure to IL6^{7, 11}. We note the majority of the axis dysregulated through IL6 (*Ptpn11*, *Morrbid*, *Bcl2l11*, *Bcl2*, *Casp1*) is not differentially expressed in the context of IL1 β exposure, suggesting a different mechanism predominates⁷. In addition, *Caiado* et al. has recently shown that IL1 α promotes CH through likewise mechanisms as IL1 β ⁴⁶. However, it is noteworthy that chronic IL1 α increased proliferative signatures and reduced *in vitro* self-renewal capacity of *Tet2*^{+/-} cells, whereas IL1 β reduced proliferative signatures with increased self-renewal capacity *Tet2*-KO cells relative to WT counterparts. Whether this difference is resultant from homozygous versus heterozygous loss of *Tet2*, dosage dependencies, or differences between the spatially non-redundant roles of IL1 β and IL1 α remains to be investigated⁴⁷. This suggests various cytokines may differ in the mechanism by which they promote the expansion of *TET2*-mutated cells. Combined blocking of multiple cytokines or their downstream mechanisms may be necessary to fully abrogate clonal expansion.”

4. As stated earlier, it is important to specify in the discussion that CHIP in human is more complex and that many other factors other than inflammation may play a role. (This not precluding clinical trial in appropriate subjects with anti-inflammation intervention, there is one that has already started in CCUS using anakinra).

Response: We agree with the importance of clarifying the complexity of CHIP in humans and have added a sentence within our conclusion clarifying that findings may not perfectly translate to human CHIP because of the relative complexity of the disease, but that the findings support additional human mechanistic investigation on page 22.

“Cumulatively, our work highlights the mechanism and therapeutic potential of inhibiting IL1 β signaling for suppressing or delaying CH-associated expansion, which may apply to the more complex multifactorial CHIP-associated premalignant myeloid expansion observed within patients.”

Reviewer #2 (Remarks to the Author):

General comment

In their study, McClatchy J. et al. address the molecular mechanisms underlying IL-1B-mediated Tet2-KO clonal hematopoietic expansion. The research question builds on recent studies, highlighting inflammation and particularly IL-1 as a driver of Tet2-mutant leukemogenesis. In particular, a recent screen in up to 22092 individuals with clonal hematopoiesis revealed that increased IL-1B circulating levels associate significantly and exclusively with TET2 mutant clonal hematopoiesis (Bick AG et al. Nature 2020). Moreover, mouse studies have implicated increased IL-1 levels in aged animals as a driver of Tet2-mutant clonal expansion (Caiado F, et al. Blood 2022; Burns SS, et al. Leukemia 2022). While the observation that IL-1 drives Tet2-mutant clonal expansion is already well established and not novel, the precise molecular underpinnings are not addressed yet, and they are of key relevance to the field.

To address this, the authors used different bone marrow (BM) chimera setups carrying WT and Tet2KO hematopoiesis and observed that prolonged IL-1B exposure leads to expansion of the Tet2KO myeloid compartment (particularly CD11b+Gr1high subset) in BM, blood and spleen (Sp). Additionally, IL-1 exposure led to a general but heterogeneous expansion of the BM and Sp HSPC compartment (MPP2 in BM and LT-HSCs/MPP2-4/CMPs/GMPs/MEPs in Sp). Functional analysis of WT or Tet2KO LSKs exposed to vehicle or IL-1 revealed increased re-plating capacity (an in vitro measurement of self-renewal) of IL-1 Tet2KO LSKs. To identify the molecular mechanisms underlying this IL-1 mediated expansion, the authors performed sc-RNA seq on lineage-depleted BM cells from WT and Tet2KO mice treated with vehicle or IL-1 for 5 weeks. Sc-RNA seq data revealed increased expansion of IL-1 treated Tet2KO granulocyte/monocyte progenitor cluster. Focusing on the HSCs cluster, the authors identify increased transcriptional signatures associated with chronic stress, self-renewal and myeloid priming, and decreased signatures associated with proliferation and lymphopoiesis in IL-1 treated Tet2KO samples. Whole genome bisulfite sequencing in purified LSKs, CMPs and GMs from vehicle or IL-1 treated Tet2KO or WT mice revealed that Tet2KO resist IL-1-driven methylation reduction of lineage-specific enhancer regions and terminal differentiation promoting transcription factor binding sites. This data suggests that failure of Tet2KO cells to demethylate these sites upon IL-1 exposure may account for their enhanced self-renewal capacity. Having established that IL-1 drives Tet2KO clonal expansion and myeloid bias, the authors performed genetic ablation or pharmacological inhibition (using IL1R1 antagonist anakinra) of Il1r1 in WT or Tet2 KO mice and observed a delayed myeloid and progenitor expansion, splenomegaly and overall survival of Tet2;Il1r1KO mice. To validate the findings in human cells, the authors deleted Tet2 from human progenitors and confirmed their IL-1 induced expansion in vitro.

Overall, this is a relevant and generally well-performed study, confirming previous findings on IL-1 as a targetable driver of Tet2-mutant clonal expansion and leukemogenesis. The novelty of the study lies in describing the underlying molecular aspects (transcriptional and epigenetic) by which IL-1 drives Tet2-KO myeloid expansion.

Response: We thank the reviewer for recognizing that our study is relevant and advanced the mechanistic understanding of IL-1-mediated clonal expansion.

Major comments (in random order)

1. The BM chimera approach used in figure 1 has multiple limitations that need to be addressed:

a) The treatment with IL-1 starting at 3 weeks post transplantation is likely somewhat too early, since multilineage repopulation after total BM transplantation takes at least 8-12 weeks for complete reconstitution. This fact is evident when considering the frequency of PB CD11b⁺ cells in untreated WT CD45.1⁺ fraction, which is 90% at week 3 and only stabilizes to the expected 15% (PMID: 18544662) by week 10. The experiment should be performed starting the IL-1B treatment the earliest by week 8 post transplantation to avoid confounding inflammatory signals derived from irradiation.

Response: We agree that the initial myeloid frequency of 90% is due to the repopulation of the marrow (Figure 1b).

1. We have added a cautionary statement in the Results section about the shortcomings of treating mice 3 weeks post transplantation by adding the following sentence: on page 5

“Myeloid (CD45⁺CD11b⁺) cell frequency was elevated (~90% of the PB) 3 weeks post-transplantation, as lineage reconstitution is likely incomplete at that time due to pre-conditioning of recipient mice with irradiation. However, myeloid bias remained elevated in *Tet2*-KO relative to WT to a greater extent in mice treated with IL1 β for 15 weeks, while myeloid frequency normalized in vehicle-treated mice (Fig. 1b).”

2. We also note that the experiment conducted within the Rosa *Tet2* transplantation model involved only small doses of IL1 β (6 total days of exposure) prior to week 9 post-transplantation. Despite this, there was a significant expansion of *shTet2* during the robust chronic treatment of IL-1 β spanning 4 weeks, which began 11 weeks post-transplantation (Fig. 1e). Notably removal of doxycycline and IL1 β reversed *shTet2* and myeloid expansion all the way out to week 10 post-transplantation, at which point doxycycline and IL-1 β induction drove the profound expansion and myeloid bias in *shTet2* not observed in vehicle-treated mice (Fig. 1f). **We have also clarified within the figure legends details related to the treatment of mice with IL1 β in the *shTet2* induction study to highlight point 1 above. The addition to the legend of Figure 1 is shown below.**

“Arrows indicate the time relative to the initial treatment (week: 0, 3, 6, 8) and duration of periods (days: 2, 4, 7, and 28) during which daily treatment with IL1 β or vehicle occurred (n = 4-5 mice/group). The green color indicates the time and duration of doxycycline treatment.”

3. To address the concerns related to irradiation, we note that mice were also treated in a non-transplantation setting with IL1 β and the identified transcriptional self-renewal signatures occurred outside of a competitive transplantation setting. Furthermore, without transplantation neutrophil expansion and skewed Ly6c^{hi} to Ly6c^{lo} monocyte/macrophage ratios were both observed (Supplementary Fig. 4e, f). Our genetic IL1R1 knockdown with *Tet2*-KO study is performed in a non-transplantation setting also supporting the overall conclusion of the manuscript (Fig 5).
4. The novelty of our work lies in demonstrating the epigenetic and transcriptomic mechanism of the competitive advantage conferred by the addition of IL1 β to *Tet2*-KO clonal expansion. This advantage, compared to the wild-type, is still observed up to 15 weeks of treatment (18 weeks post-transplantation). While pre-conditioning may contribute to some degree of expansion, the main point remains that IL1 β exaggerates the degree of this expansion.

We also agree that the study proposed by the reviewer would be informative and will shape future studies we perform.

b) The fact that WT and Tet2KO are in different CD45 isoforms might explain in part the excessive expansion of Tet2KO (even in the absence of IL-1) and might confound the effect of IL-1 β . Indeed, a 25% reduction in homing efficiency, 3.8-fold reduction in transplantable long-term hematopoietic stem cells, a 5-fold reduction in LT-HSCs capable of 24-hour homing, and a cell-intrinsic engraftment defect of 30% to 50% was reported in B6.SJL-derived bone marrow cells (CD45.1) relative to B6-derived cells+ total BM (CD45.2) (PMID: 19901263). The authors should add two control groups with WT CD45.2 competing with WT CD45.1 in the presence of vehicle or IL-1 β .

Response: We agree that differences within the competitive fitness of CD45.1 relative to CD45.2 cells are of concern. To directly address this comment, we performed a bone marrow (BM) transplant in which CD45.2 (WT or *Tet2*-KO) BM cells were transplanted in a 1 to 3 ratio with WT CD45.1 counterparts. WT CD45.2 cells maintained close to the original 1 to 3 ratio (~20%) whereas *Tet2*-KO cells by week 4 and 8 post transplantation are at ~50% (data shown below). In further support of this, the *Lys-Cre* transplant (Supplementary Fig. 6, formerly Supplementary Fig. 4), is also a CD45.2 vs CD45.1 transplant but does not show CD45.2 expansion overtime. Additional support is observed *in vitro*, by colony formation assay (Figure 2D) and in the findings of Moran-Crusio et al. 2011 (PMID: 21723200), which showed only a trend towards the expansion of WT CD45.2 over CD45.1, with *Tet2*-KO cells having superior fitness overtime. Cumulatively, we conclude WT CD45.2 alone does not appear to drive the degree of expansion relative to WT CD45.1 cells as observed in *Tet2*-KO CD45.2 cells. The data below is included in Supplementary Fig. 3f.

Supplemental Figure 3f: Lineage-depleted BM from CD45.2 (WT or Tet2-KO) cells and WT CD45.1 cells were transplanted into lethally irradiated WT CD45.1/2 recipients in a 1 to 3 ratio and analyzed 4 and 8 weeks post-transplantation by flow cytometry (n = 8 mice/group).

2. A key aspect of the study is the expansion of Tet2KO HSPCs upon IL-1B treatment. However, the authors do not show this in the competitive chimera setup and rather choose to show this in the more complex experimental setup using the doxycycline-inducible shRNA targeting Tet2. Given the importance of this observation, they should show the BM HSPC chimerism relative to figure 1a. Moreover, there is some conflicting data concerning the expansion of the HSPC compartment when using FACS (figure 2) and scRNA-seq. This discrepancy needs to be clarified.

Response: We have included the flow cytometric analysis of the competitive chimera setup for the reviewer to examine (included below). We note that in this initial experiment that was performed towards the beginning of our studies, markers for CD48 were not included, and thus fully distinguishing HSPCs subsets is not feasible. This is the initial reason for concluding only off the more robust doxycycline inducible setting. Our overall conclusions are similar from this data though we note they differ slightly as the gates cannot define all populations. For example: All HSPCs subsets are elevated in the spleen, with a greater degree of elevation in frequency in the MPP2/MPP3/HSC inclusive gate relative to the MPP4 inclusive gate. Within the BM, the MPP4 inclusive population is reduced in frequency in *Tet2*-KO treated with IL1 β compared to *Tet2*-KO treated with vehicle, while all other populations were comparable. The frequency of GMPs to MEPs is elevated in the BM of *Tet2*-KO treated with IL1 β . A figure legend is provided below to further clarify the findings.

Figure (Review only). HSPC expansion in *Tet2*-KO mice compared to WT is enhanced by IL1 β stimulation.

Lineage-depleted BM cells from WT CD45.1 and *Tet2*-KO CD45.2 mice were transplanted into lethally irradiated WT CD45.1/2 mice and after 3 weeks treated with IL1 β (500 ng/mouse/day) or vehicle daily. (a-b) Flow cytometric analysis of the bone marrow and spleen for the percentage of live CD45⁺ cells which are (a) the LT-HSC/MPP2 inclusive subset, the MPP2/MPP3/HSC inclusive subset, the MPP4 inclusive subset, (b) CMPs, GMPs, and MEPs. Error bars represent mean \pm SEM. Two-factor ANOVA was used to determine the FWER adjusted p values: *p < 0.05, **p < 0.01, ***p < 0.001, ****p < 0.0001.

Regarding the comment “there is some conflicting data concerning the expansion of the HSPC compartment when using FACS (figure 2) and scRNA-seq.” The following reasons may explain our results:

- 1) It should be noted that the data from the scRNAseq does not come from the competitive transplant mice observed in Figure 2 but from non-transplanted mice, whose data is shown in supplementary Figure 5b.
- 2) The BM from the non-transplanted mice for single cell analysis underwent magnetic selection for lineage-negative cells prior to scRNA-seq analysis, meaning it is more representative of % of lineage-negative cells as opposed to % of all live cells within marrow (which is what is being measured within Supplementary Figure 5). Because lineage

selection is not a perfect depletion and is performed individually on each sample, we still have varying representations of mature cells.

- 3) We also note that names for clusters utilized, although analogous to flow cytometric analysis defined counterparts are not defined by flow cytometry within the scRNAseq data as would be seen using Cite-Seq technique and therefore do not perfectly correspond to flow cytometry-defined populations.

To clarify this in the text, we have added that bone marrow cells were enriched for lineage-negative cells using magnetic selection on page 10

Minor comments

1. Please clarify the meaning of the IL-1B arrows in figure 1e.

Response: The figure legend has been updated to clarify this. These mice were not continuously treated with doxycycline and/or IL1 β but with intervals of both. The arrows indicate the time relative to the initial treatment (week: 0, 3, 6, 8) and duration (days: 2, 4, 7 and 28) of intervals of daily treatment with IL1 β and vehicle. E.g. an arrow up to 4 days IL1 β means that for 4 days, the mice were treated daily with IL1 β . We have included the revised section of the legend of Figure 1 below, with the new clarifying text underlined.

Figure 1: **(d-f)** Lineage-depleted BM cells from WT CD45.1⁺ and Rosa-rtTA-driven inducible *Tet2* knockdown (*shTet2*) CD45.2 mice were transplanted into lethally irradiated WT CD45.1⁺CD45.2⁺ mice, which were treated with doxycycline (dox) two weeks after transplantation and IL1 β or vehicle three weeks after transplantation for increasing intervals up to 10 weeks and then with and without doxycycline (withdrawal; w/d) for an additional two weeks, analyzed by flow cytometry. Arrows indicate the time relative to the initial treatment (week: 0, 3, 6, 8) and duration of periods (days: 2, 4, 7, and 28) during which daily treatment with IL1 β or vehicle occurred (n = 4-5 mice/group). The green color indicates the time and duration of doxycycline treatment.

2. Tet2 expression in GFP⁺ and - cells should be shown (relative to Figure 1d-e).

Response: We have performed qPCR for *Tet2* on FACS-purified cell pellets, showing that doxycycline knockdown reduces *Tet2* expression to ~20% of the original expression. The data is included below and in Supplementary Figure 3b and is referenced on page 6 of the main text.

“Doxycycline-induced knockdown of *Tet2* was monitored by GFP expression and remained reduced by ~80% at the endpoint by quantitative PCR (Supplementary Fig. 3a, b)¹⁹.”

Supplementary Fig. 3b: Lineage-depleted BM cells from wild-type (WT) CD45.1⁺ and Rosa-rtTA-driven inducible Tet2 knockdown (shTet2) mice were transplanted into lethally irradiated WT CD45.1/2 mice, which were treated with doxycycline (dox) two weeks after transplantation and IL1 β or vehicle three weeks after transplantation for increasing intervals up to 10 weeks and then with and without doxycycline (withdrawal, w/d) for an additional two weeks, analyzed by flow cytometry (n = 4-5 mice/group) (described in Fig. 1d). **(b)** qPCR of *Tet2* expression from CD45.2⁺GFP⁺ and CD45.1⁺ cells sorted from the BM of IL1 β and doxycycline-treated mice (n = 3).

3. In SuppFigure 2B the initial levels of CD11b⁺ cells seem too high (50% when in general it should be around 15-20%). Can the authors comment on this?

Response: We agree, the initial frequency is higher, but all mice given vehicle eventually normalized to the 15 – 25% observed in mice of around 12 weeks of age (e.g. week 2 onwards). The exact reason for the initial high frequency is unknown, it is possible that despite being raised in specific pathogen free housing, that mice had some form of acute exposure to a pathogen.

4. Group comparisons statistical analysis are done using independent t-testing. However, for multiple comparisons the authors should perform A-NOVA. This would make the significant differences more clear.

Response: Thank you for this suggestion, we discussed with our bioinformatician and all conclusions with multiple comparisons are now performed by a two- or three-factor ANOVA where appropriate. All figure legends, as well as the significance within all figures, have been adjusted accordingly.

5. When describing figure 2A the authors claim increased LT-HSC frequency in shTet2 upon IL-1 β stimulation compared to vehicle. However, this increase is not significant. This needs to be clearly indicated.

Response: This difference is determined significant by ANOVA and the Figure has been updated accordingly.

6. In cell cycle analysis, the authors should separate G0 from G1 fractions.

Response: This graph has been updated to include a separated G0 and G1 fraction (Supplementary Figure 7b), the text has been updated to reflect this (as well as to reflect the differences found by ANOVA testing). No significant difference is observed in G0 frequency alone. We include the below description on pages 10.

“Upon *in vivo* IL1 β administration, *Tet2*-KO LSKs trend towards lower proportion of cells in S/G2M relative to IL1 β -stimulated WT LSK cells (Supplementary Fig. 7b), suggesting a higher proportion of *Tet2*-KO LSKs maintained as G₀/G₁, although no significant difference was noted in either of those populations.”

Supplementary Figure 7b: Flow cytometry analysis of LSK cells isolated from the BM of *Tet2*-KO and WT mice treated with and without IL1 β for 5 weeks, with cell cycle proportions using KI-67 and DAPI (right, n = 4 mice/group).

7. Figure numbering needs to be carefully checked (e.g. page 9, line 200 I believe the authors are referring to figure 2d).

Response: We apologize for this mistake. All numbering has been double checked and updated.

8. Frequency of BM HSPCs populations (LT-HSCs, MPPs...) concerning anakinra treatment (relative to Figure 5e-g) should be shown.

Response: Unfortunately, compensation error for a few markers during this experiment eliminated the ability to show the frequency of LT-HSCs and MPPs.

9. Recent data by Caiado et al. (PMID: 36379023) shows that IL-1 is a driver of Tet2-mutant clonal hematopoiesis. This data is highly relevant for the current study and should be cited and discussed in context of the here presented findings.

Response: We agree the work recently published by Caiado et al. after the submission of our manuscript to Nature Communications on 2nd Feb is highly relevant and have included it as shown below on pages 19 and 22 of our discussion:

“In addition, Caiado et al. has recently shown that IL1 α promotes CH through likewise mechanisms as IL1 β ⁴⁶. However, it is noteworthy that chronic IL1 α increased proliferative signatures and reduced *in vitro* self-renewal capacity of *Tet2*^{+/-} cells, whereas IL1 β reduced proliferative signatures with increased self-renewal capacity *Tet2*-KO cells relative to WT counterparts. Whether this difference is resultant from homozygous versus heterozygous loss of *Tet2*, dosage dependencies or differences between the spatially non-redundant roles of IL1 β and IL1 α remains to be investigated⁴⁷.”

“Consistent with *Caiado et al*⁴⁶, treatment with anakinra, a clinically approved IL1 receptor antagonist, reduced the myeloid expansion of *Tet2*-KO cells.”

Reviewer #3 (Remarks to the Author):

The manuscript by McClatchy et al presents results demonstrating the relationship between IL1b and loss-of-function TET2 mutations in promoting an increased self-renewal ability of the hematopoietic stem and progenitor cells (HSPC), a myeloid-skewed hematopoiesis, and the differentiation of pro-inflammatory MHCII+ macrophages. To do so, the authors used a transplantation model using HSPCs from the Tet2-KO mouse strain developed by Aifantis I and Irvine RL (reference 6), and investigated hematopoiesis and hematopoietic progenitors upon IL1b stimulation. They also used an inducible-shTET2 mouse model to mimic the acquisition of somatic mutations in Tet2 (reference 19). The study was well conducted and results supported the conclusions of the authors. However, some issues remain to be clarified.

Response: We appreciate the reviewer highlighting the strength of our study and their encouragement.

Major comments:

- The authors deeply described the bias observed in the mature neutrophils and monocytes/macrophages subsets in spleen and bone marrow (BM) upon stimulation by IL1b. In the Tet2-KO mouse model, T-cells were reduced in Tet2-KO mice compared to WT with or without addition of IL1b whereas B-cells were reduced only in presence IL1b. What are the results concerning lymphocytes percentages and counts in the inducible mouse model? Moreover, the Tet2-KO mice showed increased percentages of lymphocytes in blood compared to WT in supplemental figure 2c. These results look contradictory with the ones showed in supplemental figure 1b and would need to be commented.

Response: Thank you for the suggestions. We have included the results from the inducible mouse model for T Cells and B Cells. Within the peripheral blood of this model, we also see T Cell reduction. We included these results on page 6 and they are included as Supplementary Figure. 3c and d.

“As observed with *Tet2* KO mice (Supplementary Fig. 2b), T cells were reduced in frequency in the PB of *shTet2* cells independent of IL1 β , whereas B cell frequency loss was IL1 β dependent (Supplementary Fig. 3c). BM T and B cell frequency was also reduced within IL1 β treated *shTet2* cells compared to vehicle, with splenic T Cells trending towards a reduction (Supplementary Fig. 3d).”

Supplementary Figure. 3c and d: Lineage-depleted BM cells from wild-type (WT) CD45.1⁺ and Rosa-rtTA-driven inducible Tet2 knockdown (*shTet2*) mice were transplanted into lethally

irradiated WT CD45.1/2 mice, which were treated with doxycycline (dox) two weeks after transplantation and IL1 β or vehicle three weeks after transplantation for increasing intervals up to 10 weeks and then with and without doxycycline (withdrawal, w/d) for an additional two weeks, analyzed by flow cytometry (n = 4-5 mice/group) (described in Fig. 1d). Arrows indicate the time relative to the initial treatment (week: 0, 3, 6, 8) and duration of periods (days: 2, 4, 7, and 28) during which daily treatment with IL1 β or vehicle occurred (n = 4-5 mice/group). The green color indicates the time and duration of doxycycline treatment. (c) The percentage of WT CD45.1 or *shTet2* CD45.2 cells which are B (CD45⁺CD11b⁻B220⁺) or T (CD45⁺CD11b⁻CD3⁺) cells in the PB, (d) BM and Spleen.

In regards to CBC results (Supplementary Figure 4d, formerly Supplementary Figure 2c), we now include additional graphs showing the T and B cell frequency in the PB (Supplementary Figure 4c). IL1 β dependent loss in T cell frequency is also observed in this experiment, however, IL1 β independent *Tet2*-KO reduction in T cells is not observed. This suggests IL1 β independent *Tet2*-KO reduction in T cell frequency may depend on transplantation-specific factors (inflammation, irradiation, replicative reconstitutive stress). On page 7 we have added text to reflect this:

“IL1 β dependent reduction of T and B cell frequency were observed again in *Tet2*-KO. Notably, lymphoid cell number was not reduced in the PB, suggesting lymphoid frequency reduction is a result of myeloid expansion not reduced lymphoid cell production. IL1 β independent reduction in T cell frequency was not observed in *Tet2*-KO mice without transplantation, suggesting it requires transplantation dependent factors, e.g. irradiation, replicative reconstitutive stress (Supplementary Fig. 4c, d).”

Supplementary Figure 4: Chronic IL1 β exposure enhances myeloid expansion of *Tet2*-KO cells in non-competitive murine models.

Tet2^{fl/fl} (WT) and *Vav-Cre Tet2*^{fl/fl} (*Tet2*-KO) mice were treated with IL1 β (500 ng/mouse/day) or vehicle (n = 4 mice/group) for five weeks and peripheral blood (PB), bone marrow (BM) and spleen samples were analyzed by flow cytometry. (c) The percentage of the PB CD45⁺ cells which are T Cells or B Cells. (d) Differential blood cell counts representing granulocytes, monocytes, and lymphocytes in the PB.

- In figure 3g, the authors used the scRNAseq signature based on the differentially expressed genes found in Tet2-KO vs WT HSCs at steady and pro-inflammatory states to analyze TET2 mutated AML primary samples. The choice to study primary AML is rather surprising as TET2 is not the most frequently mutated gene in this disease by contrast to myelodysplastic syndromes (MDS) (Ogawa S. Blood 2019). Moreover, because of the continuum in the progression from clonal hematopoiesis to MDS and then to secondary AML, MDS seem to be more relevant as it better corresponds to the model of study developed by the authors. Applying the signature to TET2-mutated MDS samples compared to TET2 WT would probably provide interesting results.

Response: We agree that analysis of MDS data-set would be closer to the state of clonal hematopoiesis and therefore more likely to have largely overlapping signatures. We are unable to access a database for MDS with paired transcriptional data accompanied with mutation data to perform this analysis in *TET2* mutated MDS. Therefore, we performed analysis on AML with and without *TET2*-mutations, and the results are encouraging to pursue this analysis in an MDS cohort in the future.

- The analysis of the DNA methylation in the Tet2-KO LSK compared to WT upon IL1b treatment showed hypermethylation of the enhancers, notably for lymphoid genes. The authors performed this experiment on LSKs, CMPs and GMPs. However, what is the DNA methylation landscape in the Common Lymphoid Progenitor (CLP) subset in the presence or absence of IL1b?

Response: We agree that analysis of CLP methylation would be an interesting experiment and is a fascinating question for future studies. In this study, we primarily focused on the myeloid lineage, and consequently, the CLP subset was not sorted, nor analyzed by flow cytometry in our study. It is an intriguing direction to focus on the impacts of *Tet2* mutations on the lymphoid lineage and explore the methylation landscape of this compartment. However, this question has prompted us to explore the HSC transition in pseudotime to CLPs which is repressed upon IL1 β stimulation and is included in Supplementary Figure 9b on page 13.

Supplementary Figure 9b: Cells analyzed by 10X single cell RNA (scRNA) sequencing (described in Figure 3a) were assigned pseudotime values using Monocle3. **(b)** Pseudotime as HSCs differentiate over myeloid (HSC, Prog, IMP, GMP), erythroid (HSC, Prog, MEP, CFU-E, EryB) and lymphoid (HSC, CLP) lineage trajectories.

- A recent report showed that IL1a and IL1b do not play redundant roles during immune response (Eislmayr et al Science 2022). Even more recently, an article by Caiado et al suggested a major role for IL1a in the age-driven $Tet2^{+/-}$ clonal hematopoiesis (Caiado et al Blood 2023). How do the authors reconcile their observations with the one by Caiado et al? Does IL1a collaborate with IL1b in their model?

Response: Thank you for bringing up this point. Interestingly $Tet2^{+/-}$ BM stimulated *in vitro* with IL1a exhibits reduced self-renewal capacity, but not to the extent that WT stimulated with IL1a does. In contrast, IL1β promoted self-renewal capacity in $Tet2$ -KO. The implication of these

findings and how they cooperate with the findings of (Eislmayr et al Science Advances 2022, reference number 47 below) have been expanded in the discussion (on page 19 and 22) and the exerts have been included below for convenience.

“In addition, Caiado et al. has recently shown that IL1 α promotes CH through likewise mechanisms as IL1 β ⁴⁶. However, it is noteworthy that chronic IL1 α increased proliferative signatures and reduced *in vitro* self-renewal capacity of *Tet2*^{+/-} cells, whereas IL1 β reduced proliferative signatures with increased self-renewal capacity *Tet2*-KO cells relative to WT counterparts. Whether this difference is resultant from homozygous versus heterozygous loss of *Tet2*, dosage dependencies, or differences between the spatially non-redundant roles of IL1 β and IL1 α remains to be investigated⁴⁷.”

“Consistent with *Caiado et al*⁴⁶, treatment with anakinra, a clinically approved IL1 receptor antagonist, reduced the myeloid expansion of *Tet2*-KO cells. In addition, we identified that *TET2*-edited human progenitors produced increased monocyte colonies upon IL1 β stimulation, which is blocked by IL1 antagonism which further suggests therapeutic potential.”

Minor comment:

- Pages 9-10. There is no panel f, nor g, in figure 2.

Response: We apologize for this oversight, we have corrected this error, and gone over all figure references in detail.

Nicolas DULPHY.

Reviewer #4 (Remarks to the Author):

McClatchy and colleagues investigated the effect of IL1 β signaling on HSPCs in a model system of clonal hematopoiesis (CH). To study the development of hematological malignancies in CH they used a murine TET2 loss-of-function model, as TET2 is one of the most common mutations in CH. The authors explored the changes in cell type abundancies upon TET2 KO and IL1 β stimulation using flow cytometry and could recapitulate the known expansion of HSCs and the myeloid bias of HSPCs in response to pro-inflammatory stress. By using single-cell RNA sequencing of 54,984 lineage-depleted bone marrow cells, as well as whole genome bisulfite sequencing of selected hematopoietic subpopulations, the authors supported their initial findings and further elucidated the mechanism of the observed myeloid bias. They could show that TET2 KO LSK cells had higher global methylation than WT cells and this methylation was enriched in lymphoid enhancer sites. Finally, the authors could reverse the TET2 KO and IL1 β induced phenotype by using a clinically approved IL1 receptor antagonist, thus providing a promising therapeutic approach.

The authors made a great effort to validate their findings in several different mouse models (TET2 KO mice, a doxycycline-inducible shRNA targeting TET2) and with different techniques

(Chimerism experiments, scRNA seq, WGBS). However, it is unclear how novel the findings are and how many more insights beyond existing literature (for example Caiado et al. Blood 2023) the authors gained.

Response: We thank the reviewer for noting the effort taken to validate findings through multiple models and techniques.

Major comments:

1. As stated above, besides the methylation analysis it is not clear to the reader which findings are new and valuable to the field and which findings confirm existing findings. A critical discussion of what makes this study outstanding with respect to other findings is needed.

Response: Thank you for this suggestion. Our study demonstrates numerous new findings and findings which complement previously published work with comprehensive mechanistic insights that we highlighted in the discussion. These key novel findings are listed below:

1. Elevation of a ly6c^{hi} MHCII⁺ pro-inflammatory macrophage population is characteristic of IL1 β driven inflammation. This is of particular interest given the known risk of Tet2 for cardiovascular disease and the role of macrophages within cardiovascular disease.
2. IL1 β driven inflammation biases HSPCs as early as MPP3s towards the myeloid lineage.
3. LT-HSCs are expanded by IL1 β production, a finding which has recently been supported by the findings with IL1 α (Caiado et al Blood 2023) and Il1r1 knockout (burns et al Leukemia 2022) in CH, we have expanded the discussion to clarify how the mechanism may differ between IL1 β and IL1 α (see new key points from the discussion below and on page 19 in main text).
4. Self-renewal gene signatures are specifically promoted by the addition of IL1 β , whereas IL1 α reduced these signatures.
5. A core pro-inflammatory gene signature was reported which was specific to *Tet2*-KO HSCs upon IL1 β and was also observed in human AML with a *TET2* mutations versus AML without *TET2* mutations. We have expanded the discussion to include findings from more extensive analyses outside of the HSC cluster (see expanded exert from the discussion below and on page 20 in main text).
6. This is the first time identifying IL1 β specific loss of methylation in WT cells and its prevention in *Tet2*-KO and specific motifs corresponding with differentiation in HSCs and with cell fate.
7. Il1r1 depletion delayed disease progression and trended towards increased survival in aging *Tet2*-KO mice which has not been reported previously.
8. Il1r1 antagonism targeting human *TET2*-mutated cells counteracting IL1 β stimulation has not been reported.

“In addition, *Caiado et al.* has recently shown that IL1 α promotes CH through likewise mechanisms as IL1 β ⁴⁶. However, it is noteworthy that chronic IL1 α increased proliferative signatures and reduced *in vitro* self-renewal capacity of *Tet2*^{+/-} cells, whereas IL1 β reduced proliferative signatures with increased self-renewal capacity *Tet2*-KO cells relative to WT counterparts. Whether this difference is resultant from homozygous versus heterozygous loss of

Tet2, dosage dependencies, or differences between the spatially non-redundant roles of IL1 β and IL1 α remains to be investigated⁴⁷. This suggests various cytokines may differ in the mechanism by which they promote the expansion of *TET2*-mutated cells. Combined blocking of multiple cytokines or their downstream mechanisms may be necessary to fully abrogate clonal expansion.”

“Further, we identified a strong enrichment of the transcriptional signature specific to old HSCs²⁶ and AP-1 members (*Jun*, *Jund*, *Fos*, *Fosb*) which are downstream of IL1 β and other pro-inflammatory signaling at steady state⁴⁸ and were exaggerated by IL1 β stimulation in *Tet2*-KO HSCs relative to WT, suggesting *Tet2*-KO cells are primed to respond to pro-inflammatory stimulation. These pro-inflammatory genes were also observed in *TET2*-mutated versus *TET2* WT AML, suggesting human relevance, and they were specific to the HSC differentiation state, showing they are most pronounced in the population in which selective pressures are most relevant.”

2. For the scRNA seq part, the authors showed gene expression of marker genes for individual cells from each cluster, the frequency of cell types per experimental condition, and a combined UMAP. However, metadata such as the number of sequenced cells per condition and animal is missing. In addition, the analysis of the scRNA seq is not very elaborate and focuses mainly on DEG analysis in HSCs. It would strengthen the point of the paper if the author would also include an analysis of subpopulation as they did in figure 4. Other analysis such as trajectory analysis of the single conditions could be helpful to visualize the observed myeloid bias.

Response: A metadata table showing the number of sequenced cells per condition and per mouse has been provided in Supplementary Table 1.

In regards to showing DEGs outside of HSCs, we have taken the reviewer's advice and shown a heatmap of gene expression for Progenitors, IMP, and GMP clusters in Figure 3f (included below) analogous to those visualized in the methylation analysis in Figure 4c. Our data show that corresponding with LSK-specific hypermethylation of TFs driving terminal differentiation, DEGs identified by ontologies for “Self Renewal only” were only significantly downregulated within *Tet2*-KO HSCs given IL1 β . Statements describing the cell cluster identity specificity of DEGs is included in the results section on pages 11 and 12.

“Interestingly, in *Tet2*-KO relative to WT under IL1 β stimulation, these inflammation and self-renewal associated DEGs are specific to the HSC cluster (Fig. 3f).”

“*Tet2*-KO HSCs relative to WT counterparts showed predominant upregulation of genes associated with “self-renewal only” defined by GSEA analysis whereas genes associated with inflammation and self-renewal were identified in both vehicle and IL1 β -stimulated cells (Fig. 3e, 3f).”

Figure 3f: WT and *Tet2*-KO mice were treated for 5 weeks with and without IL1 β (n = 4 mice/group). BM cells were harvested and lineage-negative cells were enriched by magnetic selection. This enrichment increases the representation of progenitors but reduces the percentage of differentiated cells. Cells were analyzed by 10X single cell RNA (scRNA) sequencing. (f) Heatmap of upregulated DEGs in *Tet2*-KO relative to WT HSCs with and without IL1 β stimulation for HSCs, Prog, IMPs and GMPs. The genes shown are the lead genes for enriched pathways by GSEA analysis within HSCs, categorized into “Inflammation only”, “Inflammation and Self-renewal”, or “Self-renewal only” using the GSEA subcategories assigned in Fig. 3e.

Additionally, to better visualize the observed myeloid bias, we performed pseudotime analysis of scRNAseq data and have included the data as supplementary figure 9. We identify myeloid bias coinciding with the expanded GMP population we observe in Figure 2, as described on page 13.

“Accordingly, pseudotime analysis illustrated *Tet2*-KO IL1 β treated mice exhibited increased frequency of myeloid lineage cells whereas erythroid and lymphoid lineages contracted after differentiating from HSCs and/or progenitors along pseudotime (Supplementary Fig. 9). “

Supplementary Figure 9b: Cells analyzed by 10X single cell RNA (scRNA) sequencing (described in Figure 3a) were assigned pseudotime values using Monocle3. (b) Pseudotime as HSCs differentiate over myeloid (HSC, Prog, IMP, GMP), erythroid (HSC, Prog, MEP, CFU-E, EryB) and lymphoid (HSC, CLP) lineage trajectories.

3. Supplemental figures regarding the sorting strategy of the flow cytometry analysis are missing, especially for figure 1, which could be helpful to interpret the depicted results.

Response: Thank you for pointing this out. We have included a gating scheme for figure 1 in Supplementary Fig. 1 representing flow analysis gating for the competition studies for the manuscript.

Supplementary Figure 1: Schematics for flow cytometric analysis of differentiated hematopoietic populations. (a) Representative gating for differentiated subsets for flow cytometric analysis of the BM from chimeric mice, defining: CD11b⁺Gr1^{hi} (CD11b⁺Ly6g^{hi}Ly6c^{hi}), CD11b⁺Gr1^{lo} (CD11b⁺Ly6g^{lo}Ly6c^{lo}), B and T Cells (experimental design is shown in Fig. 1a). (b) Representative gating scheme for differentiated subsets for flow cytometric analysis of the BM from chimeric recipient mice, defining: neutrophils (Neut), ly6c^{lo} monocytes/macrophages (Ly6c^{lo} Mono/Mac), ly6c^{hi} monocytes/macrophages (Ly6c^{hi} Mono/Mac), B and T Cells (experimental design is shown in Fig. 1d).

4. Furthermore, the graphical illustration of their results is often crowded (such as Figure 2c, 5c) and needs revision.

Response: We appreciate this advice. We are slightly uncertain which exact illustrations require the most revision as fig 2c and 5c were data figures, but have reduced the graphical representation within figure 2 to reduce any unnecessary populations and to keep labelling minimal.

5. The authors used a pairwise Student's two tailed t-test for analyzing most of their experiments, but often they compare more than two variables. It is not clear to the reader if the p-value was corrected for multiple testing.

Response: Thank you for this suggestion, we engaged biostatistician Mr. Andy Kaempf to discuss our data. Now statistical tests have been changed for figures 1, 2, 4, 5 and supplementary figures 2, 3, 4, 5, 7, and 11, and are largely now performed by a two- or three-factor ANOVA where appropriate. Family wise error rate (FWER) based correction for small data sizes or false discovery rate (FDR) corrections for large data statistical inquiries are now represented and it is clearly marked in the figure legends which is used.

REVIEWERS' COMMENTS

Reviewer #1 (Remarks to the Author):

I am satisfied with the authors response to my comments.

Reviewer #2 (Remarks to the Author):

The authors improved the manuscript substantially by adding data, by better explaining/clarifying some of their statements, and by appropriate discussion; all this is greatly appreciated by this reviewer.

Reviewer #3 (Remarks to the Author):

The authors have carefully addressed reviewers' comments and suggestions for revision. They included new data in the manuscript or the supplemental materials to support their conclusions. This new draft provides significant results about the role of IL1b in the context of TET2 loss-of-function, and pave the way for new studies about IL1b and the emergence of myeloid malignancies.

Reviewer #4 (Remarks to the Author):

I appreciate the efforts in addressing my comments and incorporating the suggested changes into the manuscript. The inclusion of the supplementary data such as the meta data and the gating strategies makes it easier for the readership to follow the flow of the paper and the analysis of the pseudo time is a valuable addition to the study. As a minor comment, the reference for the Monocle3 package (line 728) is still missing. Overall, the authors have addressed my comments.

REVIEWERS' COMMENTS

Reviewer #1 (Remarks to the Author):

I am satisfied with the authors response to my comments.

We thank the reviewer for their expertise and time spent improving this manuscript.

Reviewer #2 (Remarks to the Author):

The authors improved the manuscript substantially by adding data, by better explaining/clarifying some of their statements, and by appropriate discussion; all this is greatly appreciated by this reviewer.

We appreciate the time and effort the reviewer has spent improving this manuscript.

Reviewer #3 (Remarks to the Author):

The authors have carefully addressed reviewers' comments and suggestions for revision. They included new data in the manuscript or the supplemental materials to support their conclusions.

This new draft provides significant results about the role of IL1b in the context of TET2 loss-of-function, and pave the way for new studies about IL1b and the emergence of myeloid malignancies.

We thank the reviewer for their insightful comments which have bettered our manuscript.

Reviewer #4 (Remarks to the Author):

I appreciate the efforts in addressing my comments and incorporating the suggested changes into the manuscript. The inclusion of the supplementary data such as the meta data and the gating strategies makes it easier for the readership to follow the flow of the paper and the analysis of the pseudo time is a valuable addition to the study. As a minor comment, the reference for the Monocle3 package (line 728) is still missing. Overall, the authors have addressed my comments.

We have updated the missing reference for Monocle3, we are thankful for the reviewer's thorough examination of our manuscript, it has led to a better document.